# Disruption of glycolytic flux is a signal for inflammasome signaling and pyroptotic cell death

Laura E Sanman[1], Yu Qian[2], Nicholas A Eisele[3], Tessie M Ng[3], Wouter A van der Linden[4], Denise M Monack[3], Eranthie Weerapana[2], Matthew Bogyo[1,3,4]*

[1]Department of Chemical and Systems Biology, Stanford University School of Medicine, Stanford, United States; [2]Department of Chemistry, Boston College, Chestnut Hill, United States; [3]Department of Microbiology and Immunology, Stanford University School of Medicine, Stanford, United States; [4]Department of Pathology, Stanford University School of Medicine, Stanford, United States

**Abstract** When innate immune cells such as macrophages are challenged with environmental stresses or infection by pathogens, they trigger the rapid assembly of multi-protein complexes called inflammasomes that are responsible for initiating pro-inflammatory responses and a form of cell death termed pyroptosis. We describe here the identification of an intracellular trigger of NLRP3-mediated inflammatory signaling, IL-1β production and pyroptosis in primed murine bone marrow-derived macrophages that is mediated by the disruption of glycolytic flux. This signal results from a drop of NADH levels and induction of mitochondrial ROS production and can be rescued by addition of products that restore NADH production. This signal is also important for host-cell response to the intracellular pathogen *Salmonella typhimurium*, which can disrupt metabolism by uptake of host-cell glucose. These results reveal an important inflammatory signaling network used by immune cells to sense metabolic dysfunction or infection by intracellular pathogens.

*For correspondence: mbogyo@stanford.edu

**Competing interests:** The authors declare that no competing interests exist.

## Introduction

Inflammation is an immunological process required for an organized response to infection, injury, and stress. Because excessive inflammation can be damaging, its initiation is highly regulated. Innate immune cells such as macrophages have evolved sensors of pathogens and homeostatic perturbations which, when activated, induce an immune response (*Medzhitov, 2008*). Amongst these sensors are Nod-like receptors (NLRs), which are activated in response to a diverse set of pathogen-associated molecular patterns (PAMPs) and danger-associated molecular patterns (DAMPs). Activated NLR proteins recruit and facilitate activation of the protease caspase-1 either directly, through caspase activation and recruitment domain (CARD) interactions, or indirectly, through the adaptor apoptosis-associated speck-like protein containing a CARD (ASC; also known as *Pycard*). The resulting macromolecular complex is referred to as the inflammasome (*Lamkanfi and Dixit, 2014*). The inactive precursor of the cytokine interleukin-1β (pro-IL-1β) is also recruited to the inflammasome complex, where proteolysis by caspase-1 induces activation and secretion of the bioactive cytokine, further promoting inflammation. In addition to cytokine maturation, inflammasome formation and caspase activation are associated with a pro-inflammatory form of cell death termed pyroptosis (*Fink and Cookson, 2006*). This form of cell death results in lytic release of cytosolic contents and

**eLife digest** Cells of the innate immune system, such as macrophages, are the body's first line of defense against infection. Macrophages can sense a wide variety of danger signals associated with the presence of infectious microbes, and some of these signals cause macrophages to form protein complexes called inflammasomes inside the cell. Inflammasomes produce molecules that stimulate inflammation and trigger the death of the macrophage. This attracts other immune cells to the infection site to help combat the source of danger.

Inflammasome complexes form around an activated receptor molecule called NLRP3. NLRP3 is activated by a range of danger signals, including those produced by *Salmonella* bacteria. However, the sequence of events that leads to NLRP3 activation is still not well understood.

Sanman et al. have now identified a small molecule that unexpectedly causes the formation of inflammasomes via NLRP3 and so triggers the death of macrophages. Further investigation revealed that this molecule disrupts glycolysis, a process macrophages use to produce energy. The energy imbalance caused by disrupting glycolysis triggers a stress response in macrophages, which ultimately activates the NLRP3 receptor and hence the inflammasome.

Sanman et al. then found that *Salmonella* bacteria also activate the inflammasome by disrupting glycolysis when they invade macrophages. This occurs because the bacteria use up the macrophage's supply of glycolysis precursor molecules. Replenishing the macrophage with products of glycolysis restored partial energy production and prevented the inflammasome from being activated.

Overall, Sanman et al. have identified a previously unknown trigger of inflammation and cell death in macrophages whereby cells can respond to infectious bacteria by sensing a change in energy levels. A next step will be to define the signaling molecules that activate NLRP3 to trigger the construction of the inflammasome. Sanman et al. also hope to uncover other infections and diseases where changes in energy balance might trigger inflammation and cell death.

other pro-inflammatory factors such as interleukin-1α and high-mobility group protein B1 (HMGB1), which are potent inducers of inflammation (*Medzhitov, 2008*; *Croker et al., 2014*).

Diverse activation signals have been reported as triggers of NLR signaling. For example, the NLR AIM2 is activated by cytosolic double-stranded DNA (*Lamkanfi and Dixit, 2014*; *Fernandes-Alnemri et al., 2009*; *Hornung et al., 2009*; *Bürckstümmer et al., 2009*), a structural feature associated with infections with pathogens and not found in healthy host cells (*Fink and Cookson, 2006*; *Hornung et al., 2009*; *Jones et al., 2010*). The NLR NLRP3 is a sensor of a wide variety of PAMPs and DAMPs but the unifying mechanism of its disparate activators is not understood (*Sutterwala et al., 2014*). Furthermore, while the NLRP3 signaling pathway can be activated by a variety of both gram-positive and gram-negative bacteria, the mechanism by which these pathogens induce inflammasome signaling through this receptor is often unclear (*Storek and Monack, 2015*). Specifically, effective defense against *Salmonella typhimurium (S. typhimurium)* requires NLRP3 (*Broz et al., 2010*), yet the mechanism by which the pathogen activates this pathway remains unknown.

Here, we report a small molecule, GB111-NH$_2$, that induces NLRP3 inflammasome formation, caspase-1 activation, IL-1β secretion, and pyroptotic cell death in bone marrow-derived macrophages (BMDM). Using chemical proteomics, we identify the glycolytic enzymes GAPDH and α-enolase as the phenotypically relevant targets of this molecule. Facilitating TCA metabolism downstream of glycolysis by addition of pyruvate or succinate blocked the effects of the compound. We find that *S. typhimurium* infection, like direct chemical inhibition of the glycolytic enzymes, reduced glycolytic flux and that restoring metabolism downstream of glycolysis also prevented *S. typhimurium*-induced inflammasome formation, IL-1β secretion, and pyroptosis. We find that glycolytic disruption induced by either the small molecules or *S. typhimurium* infection impaired NADH production, resulting in the formation of mitochondrial ROS that were essential for NLRP3 inflammasome activation. Therefore, disruption of glycolytic flux is a biologically relevant trigger of NLRP3 inflammasome activation

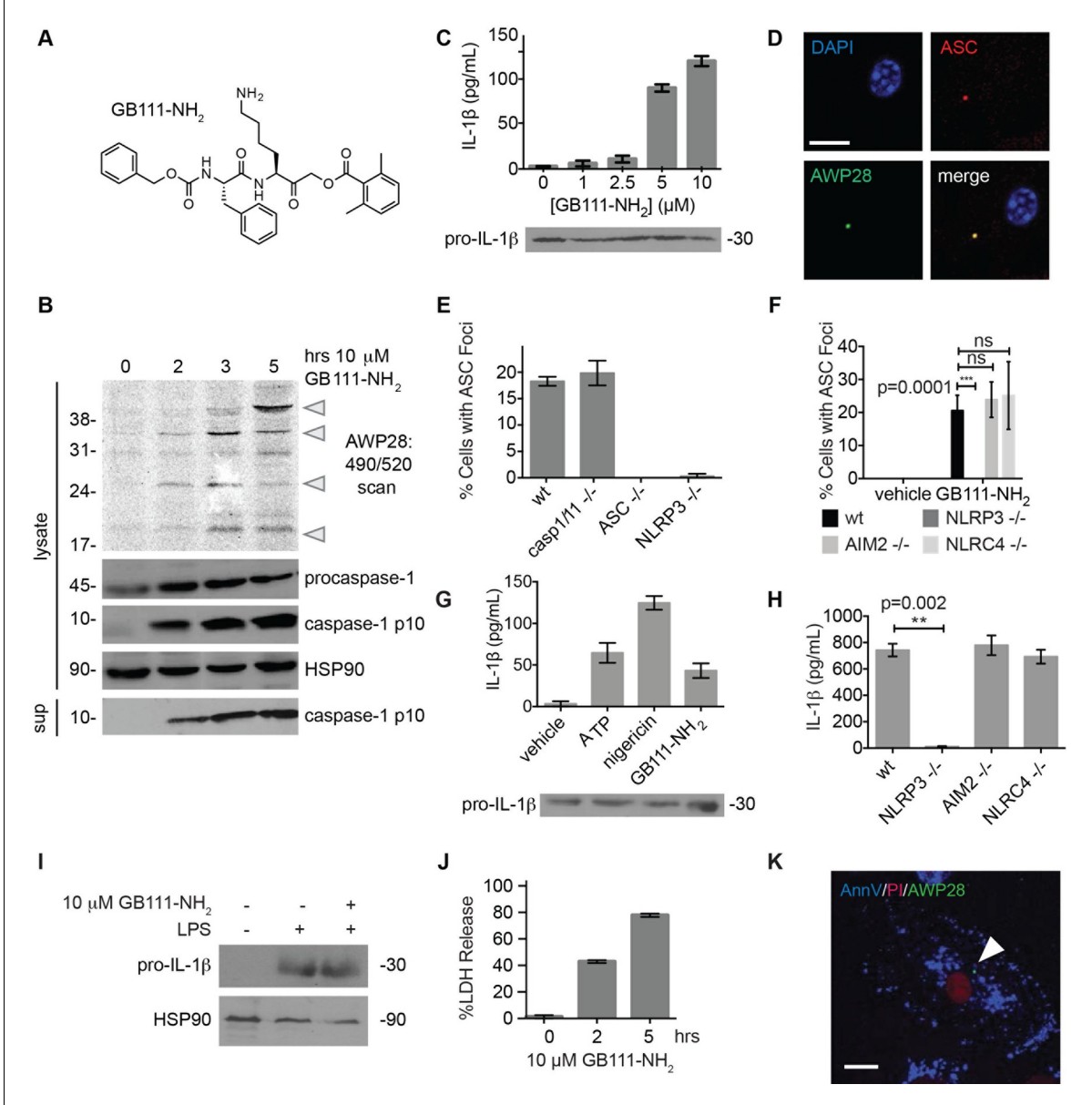

**Figure 1.** Identification of the NLRP3 inflammasome activator GB111-NH$_2$. (**A**) Structure of GB111-NH$_2$. (**B**) Western blot and activity-based probe analysis of caspase-1 activation. BMDM primed with 100 ng/mL LPS for 3 hr were then treated with GB111-NH$_2$. Intact cells were labeled with the caspase-1 probe AWP28 (1 μM) for the last hour before lysate harvest. Whole cell lysates were separated by SDS-PAGE. AWP28 labeling was analyzed by fluorescence scan and caspase-1 processing analyzed by western blot. Gray arrowheads indicate active forms of caspase-1 labeled by AWP28. HSP90 serves as loading control. (**C**) LPS-primed BMDM were treated with the indicated concentrations of GB111-NH$_2$ for 2 hr. Supernatants were analyzed by ELISA. (**D**) LPS-primed BMDM were treated with 10 μM GB111-NH$_2$ for 2 hr, labeled with AWP28, fixed, stained for ASC and DAPI, and visualized by confocal microscopy. Scale bar 10 μm. (**E-F**) BMDM of the indicated genetic backgrounds were treated with GB111-NH$_2$ as in (**D**) and inflammasome foci/nuclei quantified. At least 4 fields of view (20x objective, 0.5x zoom) were taken per condition per experiment, ~2000 cells/condition. (**G**) LPS-primed BMDM were treated with the indicated compounds (ATP: 5 mM; nigericin: 10 μM; GB111-NH$_2$: 10 μM) and supernatant analyzed by ELISA. (**H**) BMDM of the indicated genetic backgrounds were treated as in (**D**) and supernatant analyzed by ELISA. (**I**) BMDM were primed with LPS or vehicle, then treated with GB111-NH$_2$ for 2 hr. Whole cell lysates were separated by SDS-PAGE, blotted for pro-IL-1β, stripped, and reblotted for HSP90. (**J**) Cell death in LPS-primed, GB111-NH$_2$-treated BMDM was analyzed by LDH release. (**K**) LPS-primed BMDM were treated with 10 μM GB111-NH$_2$ for 2 hr, labeled with AWP28, stained for Annexin V (AnnV) and propidium iodide (PI), and visualized by microscopy. White arrowhead indicates AWP28 focus. Scale bar 10 μm. In all cases, data are representative of at least n=3 experiments and error bars indicate mean +/-sd of technical triplicate. Statistical significance was analyzed using an unpaired, two-tailed t test.

The following figure supplements are available for figure 1:

*Figure 1 continued on next page*

*Figure 1 continued*

**Figure supplement 1.** Secreted IL-1β is the bioactive, p17 form.

**Figure supplement 2.** GB111-NH$_2$ does not impair secretion of TNF-α and dose-dependently reduces IL-6 secretion.

that is mediated by mitochondrial redox changes, revealing a mechanistic link between cellular metabolism and initiation of inflammation.

## Results

### Identification of a small molecule activator of inflammasome formation and pyroptosis

While screening peptide-based compounds for their effects on inflammasome signaling, we identified one compound, GB111-NH$_2$ (*Blum et al., 2005*; *Verdoes et al., 2012*) (*Figure 1A*), that was sufficient to induce caspase-1 activation in LPS-primed bone marrow-derived macrophages. We measured caspase-1 activation by monitoring conversion of procaspase-1 to the mature p10 form by Western blot and, in parallel, by labeling BMDM with the caspase-1-selective activity-based probe (ABP), AWP28 (*Puri et al., 2012*) (*Figure 1B*). In addition to producing active caspase-1, we found that GB111-NH$_2$-treated BMDMs secreted the cytokine IL-1β in a dose-dependent manner (*Figure 1C*). Western blot analysis confirmed that secreted IL-1β was primarily the bioactive p17 form (*Figure 1—figure supplement 1*) that is generated by active caspase-1.

By fluorescence microscopy, we observed formation of foci containing the inflammasome adaptor ASC and active caspase-1 in GB111-NH$_2$-treated BMDMs (*Figure 1D*). Formation of these foci was dependent on NLRP3 and ASC but not caspase-1, caspase-11, NLRC4, or AIM2 (*Figure 1E–F*). We observed that GB111-NH$_2$ induced a similar level of IL-1β secretion as the NLRP3 stimuli ATP and nigericin (*Figure 1G*) and that the absence of NLRP3 completely abrogated IL-1β secretion induced by GB111-NH$_2$ treatment. The absence of other NLRs, specifically NLRC4 and AIM2, had no effect on IL-1β secretion (*Figure 1H*). Taken together, these data indicate that GB111-NH$_2$ induces caspase-1 activation and IL-1β secretion solely through the NLRP3 inflammasome, acting as an activating 'Signal II' for the canonical NLRP3 pathway (*Lamkanfi and Dixit, 2014*).

In order for 'Signal II' to activate the NLRP3 inflammasome, BMDM must first be primed by a 'Signal I' such as LPS. LPS priming induces NF-κB-dependent transcription of pro-inflammatory genes such as IL-1β and inflammasome-independent secretion of pro-inflammatory cytokines such as IL-6 and TNF-α (*Lamkanfi and Dixit, 2014*). We measured lysate protein levels by Western blotting and supernatant cytokine levels by ELISA in BMDM treated as in previously described experiments; first primed for 3 hr with LPS and then treated for 2 hr with GB111-NH$_2$. We observed the appearance of pro-IL-1β upon LPS priming (*Figure 1I*) but there was no effect of GB111-NH$_2$ on either IL-1β protein levels in BMDM that had received LPS priming (*Figure 1C*, *Figure 1G*). In addition, IL-6 secretion decreased with increasing dose of GB111-NH$_2$ and TNF-α secretion was unaffected by GB111-NH$_2$ (*Figure 1—figure supplement 2*). Therefore, GB111-NH$_2$ does not have a direct effect on Signal I, but functions predominantly as a Signal II for the NLRP3 inflammasome.

Macrophages containing active inflammasome complexes often rapidly die by a pro-inflammatory process called pyroptosis (*Fink and Cookson, 2006*). We observed features of this form of cell death in GB111-NH$_2$-treated BMDM, including release of the intracellular enzyme lactate dehydrogenase (LDH) (*Figure 1J*), and foci of caspase-1 activity in propidium iodide (PI) and Annexin V (AnnV) positive cells (*Figure 1K*). These data confirm that GB111-NH$_2$ is a small molecule activator of the NLRP3 inflammasome that also triggers pyroptotic cell death.

### Identification of the phenotypically relevant targets of GB111-NH$_2$

Given that GB111-NH$_2$ is chemically distinct from other known activators of NLRP3 and easily modifiable, we wanted to use it as a tool to identify protein targets that are involved in triggering this pro-inflammatory response. To accomplish this, we first conducted a small structure-activity relationship (SAR) study in which we synthesized a series of analogs of GB111-NH$_2$ to identify compounds that

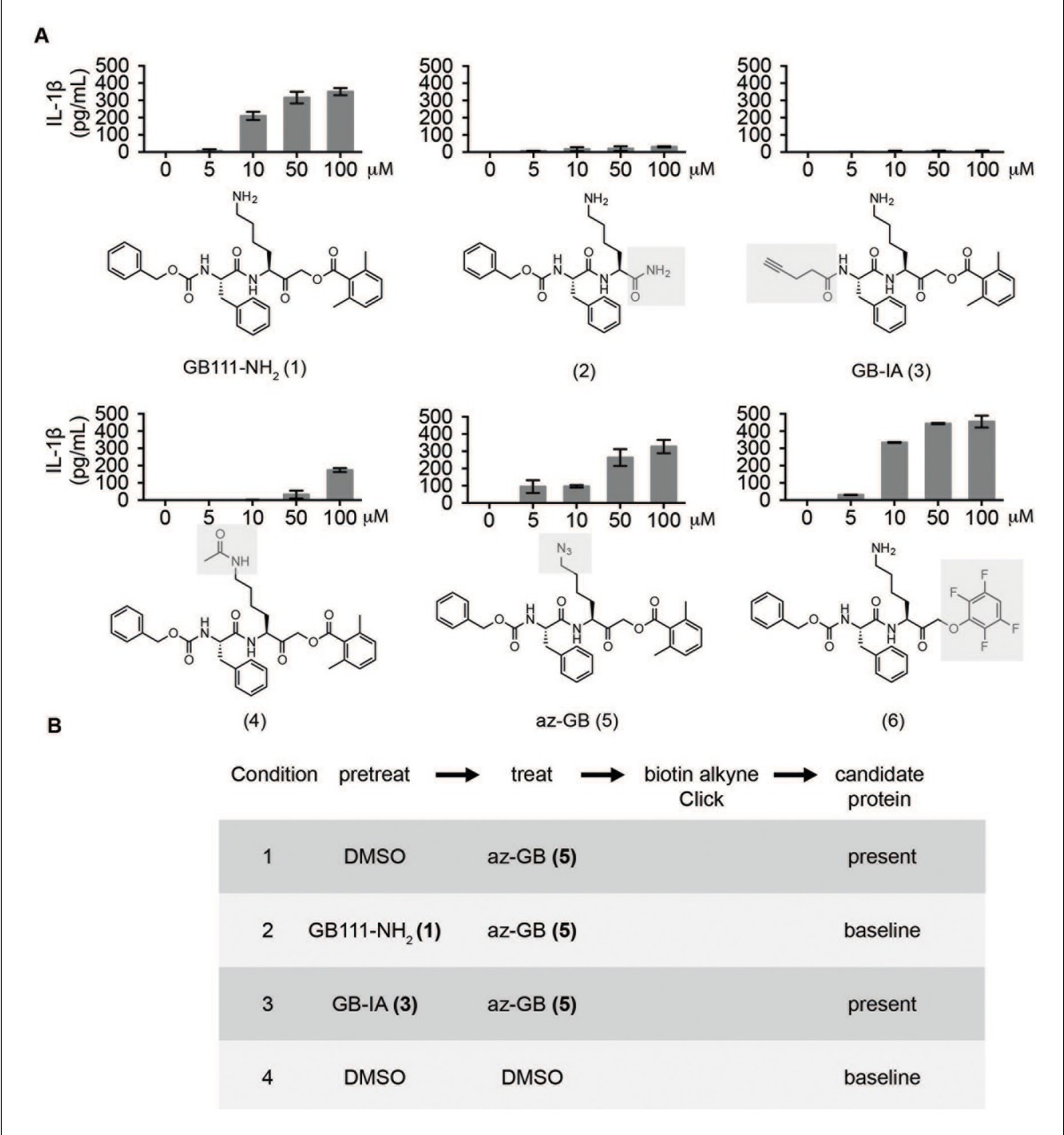

**Figure 2.** Structure-activity relationship study and identification of GB111-NH$_2$ targets. (**A**) Structures of GB111-NH$_2$ analogs with structural changes highlighted in gray. LPS-primed BMDM were treated with analogs and supernatant IL-1β measured by ELISA. A dose response is shown above each analog. (**B**) Set-up of MudPIT target identification experiment. In all cases, data are representative of n=3 experiments. Error bars indicate mean +/- sd of technical triplicate.

The following figure supplement is available for figure 2:

**Figure supplement 1.** Az-GB is a functional probe version of GB111-NH$_2$.

could be used for affinity isolation of labeled targets. We identified a number of modifications to the primary compound scaffold that resulted in loss of activity (**Figure 2A**), suggesting that the effects of the parent compound are likely dictated by affinity to specific protein targets. Importantly, our SAR efforts identified both an inactive analog (GB-IA) as well as an azide-containing analog (az-GB) that retained the inflammasome-activating ability of GB111-NH$_2$ (**Figure 2A**, **Figure 2—figure**

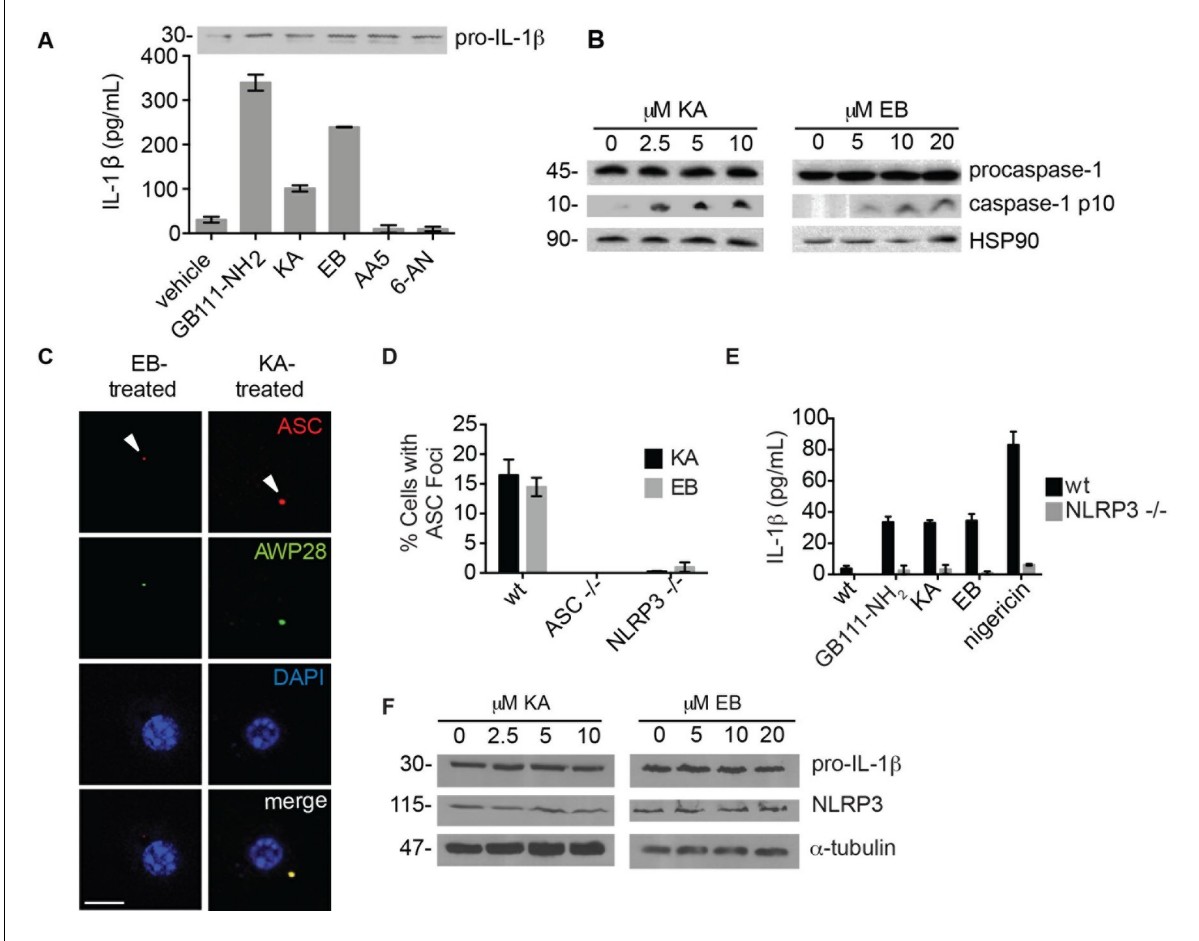

**Figure 3.** The phenotypically relevant targets of GB111-NH₂ are glycolytic enzymes. (**A**) LPS-primed BMDM were treated with the indicated compounds (GAPDH inhibitor koningic acid = KA; 10 µM, α-enolase inhibitor ENOblock = EB; 20 µM, succinate dehydrogenase inhibitor Atpenin A5 = AA5; 10 µM, 6-phosphogluconate dehydrogenase inhibitor 6-aminonicotinamide = 6-AN; 500 µM) and IL-1β secretion was analyzed by ELISA. Whole cell lysates were separated by SDS-PAGE and blotted for pro-IL-1β. (**B**) BMDMs were treated with KA and EB and cell lysates were analyzed for caspase-1 processing by western blot. HSP90 serves as loading control. (**C**) BMDM were treated as in (**B**), labeled with AWP28, fixed, stained for ASC and DAPI, and analyzed by fluorescence microscopy. Scale bar 10 µm. (**D**) LPS-primed BMDM from the indicated genetic backgrounds were treated with KA or EB for 3 hr, fixed, stained for ASC and DAPI, and analyzed by confocal microscopy. At least 4 fields of view were captured per condition, ∼2000 cells/condition/experiment. (**E**) LPS-primed BMDM of the indicated genetic backgrounds were treated with the indicated compounds (GB111-NH₂ – 10 µM for 2 hr, KA – 5 µM for 3 hr, EB – 20 µM for 3 hr, nigericin – 12.5 µM for 1 hr) and supernatant analyzed for IL-1β production by ELISA. (**F**) BMDM were treated as in (**B**). Whole cell lysates were separated by SDS-PAGE and blotted for pro-IL-1β, NLRP3, and α-tubulin.

The following figure supplements are available for figure 3:

**Figure supplement 1.** Cathepsin inhibition does not induce caspase-1 activation.

**Figure supplement 2.** Koningic acid and ENOblock induce dose-dependent IL-1β secretion.

---

supplement 1). We used this azide analog as a probe to identify potential protein targets using Click chemistry to attach a fluorescent tag (*Figure 2—figure supplement 1*) or affinity tag (biotin) to labeled target proteins. The choice of this probe does limit identification to covalent binding partners. However, because removal of the acyloxymethylketone (AOMK) electrophile from GB111-NH₂ (compound 2) resulted in loss of the ability of the compound to induce IL-1β secretion, we reasoned that the compound likely acted through covalent modification of its relevant targets.

We conducted a proteomic study in which we pre-treated BMDMs with either active or inactive analogs of GB111-NH₂, labeled with the az-GB probe, reacted the resulting lysates with alkyne-

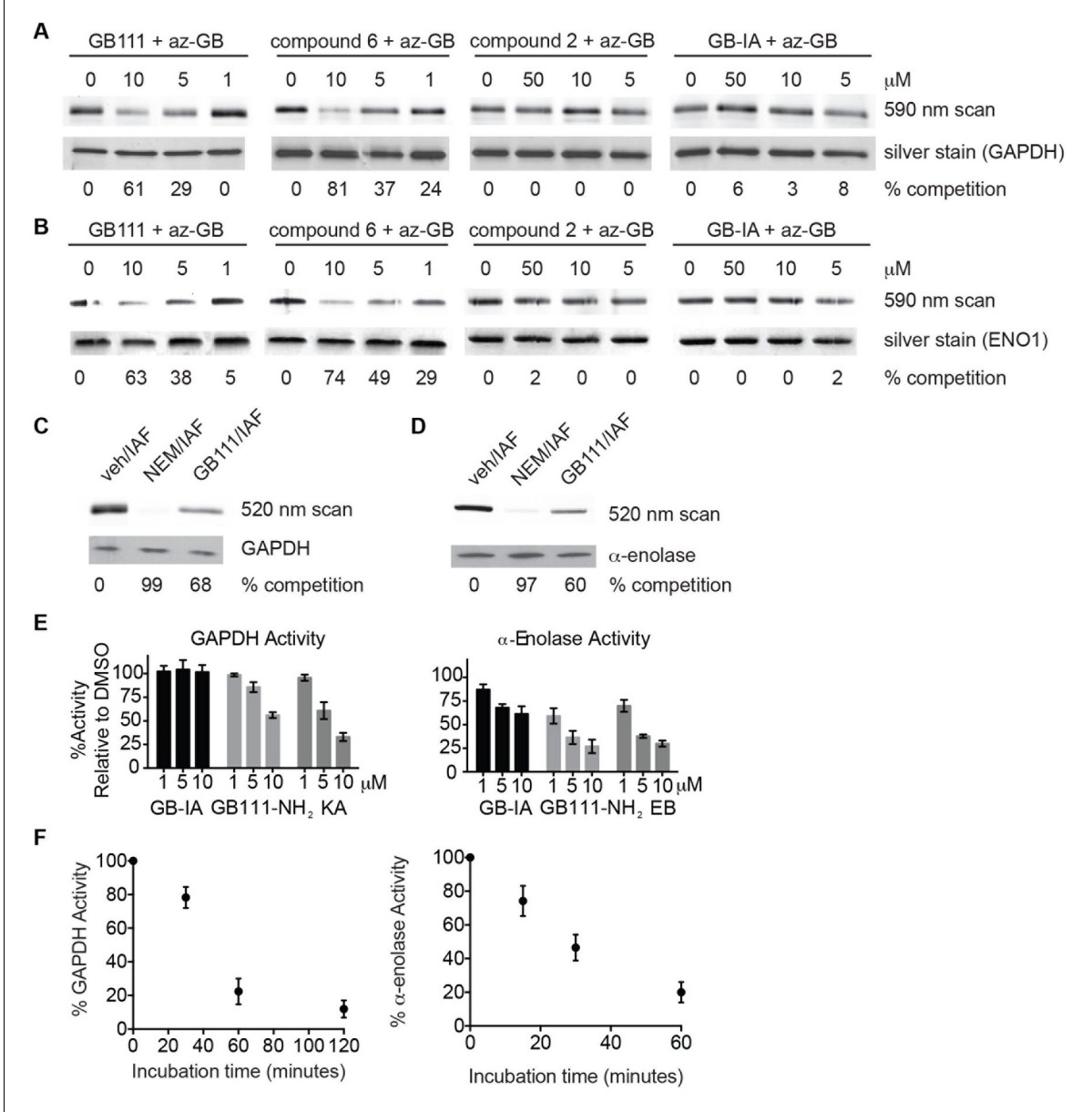

**Figure 4.** Compounds covalently bind to reactive cysteines on GAPDH and α-enolase and inhibit enzyme activity in vitro. (**A**) Recombinant human GAPDH was pretreated with GB111-NH₂ and its analogs at the indicated concentrations for 1 hr in 0.1 M Tris-HCl pH 8.0, then labeled with az-GB (50 μM) for 1 hr. Mixtures were reacted with TAMRA-alkyne, separated by SDS-PAGE, and analyzed by fluorescence scan. Gels were silver stained to assess loading. % of competition was calculated as 100-(fluor. intensity$_{cmpd+az-GB}$/fluor. intensity$_{az-GB-only}$). (**B**) Recombinant human α-enolase was labeled as described for GAPDH in (**A**). (**C**) GAPDH was incubated with NEM (5 μM), GB111-NH₂ (10 μM) or vehicle for 30 min, then labeled with iodoacetamide fluorescein (IAF; 10 μM) for 30 min. Reaction mixtures were separated by SDS-PAGE. Gels were analyzed by fluorescent scan and blotted for GAPDH to assess loading. (**D**) α-enolase was treated as described for GAPDH in (**C**) and blotted for α-enolase to assess loading. (**E**) Recombinant GAPDH and α-enolase were pretreated with inhibitors for 30 min and then enzyme activity assessed using substrate turnover assays. (**F**) GAPDH and α-enolase were incubated with GB111-NH₂ for the indicated amounts of time and then enzyme activity assessed. Data are representative of at least n=3 experiments and error bars indicate mean +/- sd of technical triplicate.

The following figure supplement is available for figure 4:

**Figure supplement 1.** az-GB binds to reactive cysteines on recombinant GAPDH and α-enolase in a manner that is dependent on enzyme activity.

biotin and identified affinity isolated targets using multidimensional protein identification technology (MudPIT) (*Weerapana et al., 2007*) (*Figure 2B*). By using active and inactive compounds in our pretreatment (GB111-NH$_2$ and GB-IA, respectively), we could identify labeled proteins that were lost by pretreatment with the active compound but not the inactive control. Employing this strategy, we obtained a short list of potentially relevant binding partners (*Supplementary files 1–3*). Interestingly, this list included proteins critical to cellular metabolism and homeostatic maintenance.

## Inhibition of glycolytic enzymes activates the NLRP3 inflammasome and induces pyroptosis

To determine which of the potential protein targets of GB111-NH$_2$ were responsible for its' phenotypic effects, we first tested whether reported selective inhibitors of several of the identified targets mimicked the effects of GB111-NH$_2$. Based on our target list (*Supplementary file 3*), we selected the compounds Atpenin A5 (*Miyadera et al., 2003*) (AA5; inhibitor of succinate dehydrogenase in the TCA cycle), 6-aminonicotinamide (*Street et al., 1997*) (6AN; inhibitor of 6-phosphogluconate dehydrogenase in the pentose phosphate pathway), koningic acid (*Endo et al., 1985*) (KA; inhibitor of GAPDH in glycolysis) and ENOblock (*Jung et al., 2013*) (EB; inhibitor of α-enolase in glycolysis). We tested these compounds for their effects on LPS-primed BMDM and found that only inhibitors of the glycolytic enzymes glyceraldehyde-3-phosphate dehydrogenase (GAPDH) and α-enolase induced IL-1β secretion, inflammasome formation, and caspase-1 processing (*Figure 3A–C*). We chose concentrations of each inhibitor based on literature descriptions of concentrations at which targets should be completely inhibited (*Miyadera et al., 2003*; *Street et al., 1997*; *Endo et al., 1985*; *Jung et al., 2013*). Mass spectrometry results demonstrated that GB111-NH$_2$ also bound to lysosomal cysteine cathepsins (*Supplementary file 2*), though not in the expected pattern. To rule out a mode of action based on cathepsin inhibition, we tested the established cathepsin inhibitors leupeptin, E-64d, and Ca074Me and found that they did not induce caspase-1 activation in BMDM (*Figure 3—figure supplement 1*). These data indicate that the phenotypically relevant targets of GB111-NH$_2$ are the glycolytic enzymes GAPDH and α-enolase.

The GAPDH and α-enolase inhibitors KA and EB failed to induce inflammasome formation in cells that lack *Pycard* or *Nlrp3* (*Figure 3D*), induced IL-1β secretion in a dose-dependent manner that was also NLRP3-dependent (*Figure 3E*, *Figure 3—figure supplement 2*), and had no effect on pro-IL-1β or NLRP3 levels in LPS-primed BMDM (*Figure 3A*, *Figure 3F*). These data demonstrate that structurally dissimilar inhibitors of either GAPDH or α-enolase activate the canonical NLRP3 inflammasome pathway similarly to GB111-NH$_2$.

To further confirm the targets of GB111-NH$_2$, we measured the ability of the parent compound and its analogs to covalently bind and inhibit the activity of the identified glycolytic enzyme targets. To test if our compounds covalently bound to GAPDH and α-enolase, we incubated recombinant GAPDH and α-enolase with our az-GB probe, used Click chemistry to attach TAMRA-alkyne to the az-GB probe, and analyzed reaction mixtures by fluorescent scanning of SDS-PAGE gels. We observed probe labeling of both GAPDH and α-enolase, indicating that the az-GB probe covalently binds to both enzymes (*Figure 4A–B*, *Figure 4—figure supplement 1*). Pretreatment with GB111-NH$_2$ blocked az-GB binding to both GAPDH and α-enolase in a dose-dependent manner, indicating that both compounds bind to similar sites on their enzyme targets. The GB111-NH$_2$ analog containing a more reactive phenoxymethylketone electrophile (compound 6), which also induced IL-1β secretion more potently in cells than GB111-NH$_2$, also blocked az-GB binding to both GAPDH and α-enolase at lower concentrations compared to GB111-NH$_2$. Importantly, the analogs that did not induce IL-1β secretion in BMDM, compound 2 (which lacks the AOMK electrophile) and GB-IA (which lacks the carboxybenzyl cap of GB111-NH$_2$), did not compete for az-GB binding to GAPDH and α-enolase. We did observe some labeling of GAPDH and α-enolase by TAMRA-alkyne independent of the az-GB probe, which is potentially due to the ability of alkynes to function as cysteine electrophiles (*Ekkebus et al., 2013*).

az-GB probe binding to both enzymes was blocked by the cysteine-alkylating compound N-ethylmaleimide (NEM), and by KA and EB (for GAPDH and α-enolase, respectively), suggesting that binding was dependent on enzyme activity and was mediated by reaction with key reactive cysteines (*Figure 4—figure supplement 1*). To further investigate the proposed covalent interaction of GB111-NH$_2$ with reactive cysteine residues in GAPDH and α-enolase, we performed competition studies with the general cysteine reative probe iodoacetamide fluorescein (IAF). IAF labeled both

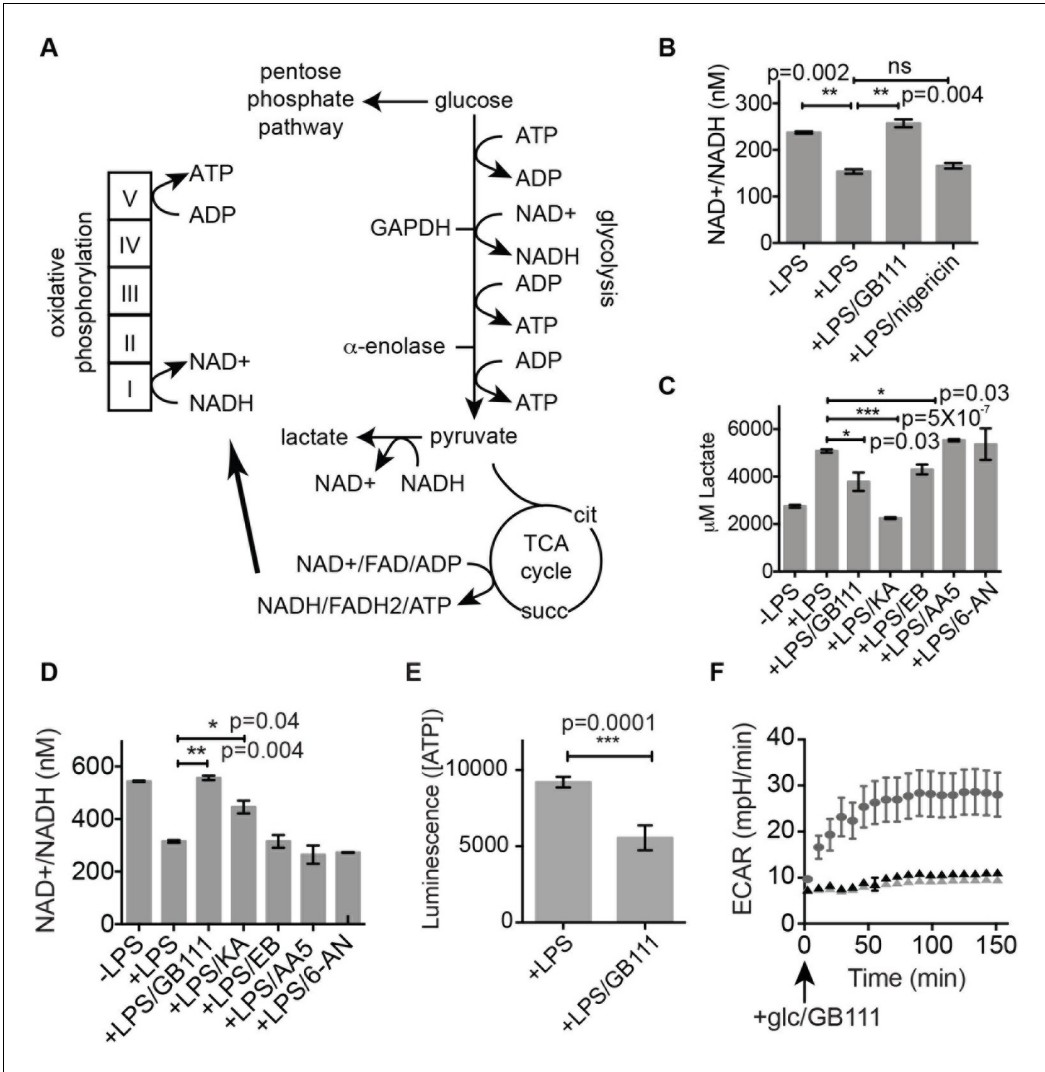

**Figure 5.** Inhibition of glycolytic flux creates a unique metabolic defect that activates the NLRP3 inflammasome. (A) Map of relevant metabolic pathways. (B) BMDM were stimulated with LPS or vehicle for 3 hr and then the indicated compounds for 2 hr, after which cytosolic NAD+/NADH ratio was measured. (C) BMDM were treated as in (B) and supernatants were analyzed for lactate production. Inhibitor concentrations are those from *Figure 3A*. (D) BMDM were stimulated with LPS or vehicle for 3 hr and then with the indicated compounds for 2 hr, after which cytosolic NAD+/NADH was measured. (E) BMDM were treated as in (B) and cytosolic ATP concentration analyzed by ATP-coupled luminescence assay. (F) ECAR was measured in BMDM upon addition of fresh glucose-containing medium. Fresh medium contained vehicle (DMSO; gray circles) or 10 μM GB111-NH$_2$ -/+ 1 mM pyruvate (black/gray triangles). Error bars represent mean +/- sd of 6 technical replicates per condition.

The following figure supplements are available for figure 5:

**Figure supplement 1.** A dose of nigericin that induces cell death with similar kinetics to GB111-NH$_2$ does not effect NADH production.

**Figure supplement 2.** 2DG does not dramatically impair glycolytic flux or induce inflammasome formation.

GAPDH and α-enolase, consistent with previous reactive cysteine profiling data demonstrating that the catalytic Cys 152 of GAPDH is highly reactive and Cys 388, an active-site proximal cysteine of α-enolase, is also reactive (*Weerapana et al., 2010*). NEM potently blocked IAF labeling, confirming that IAF was reacting with cysteine residues in GAPDH and α-enolase. Importantly, GB111-NH$_2$ also

competed for IAF labeling, indicating that it covalently binds these same reactive cysteines (*Figure 4C–D*).

Modification of the active-site cysteine of GAPDH and the active site-proximal Cys 388 of α-enolase have both been shown to potently impair enzyme activity (*Kato et al., 1992*; *Ishii and Uchida, 2004*). To confirm that binding of our compounds to these enzymes also inhibits enzyme activity, we performed substrate assays with recombinant GAPDH and α-enolase and found that GB111-NH$_2$ dose-dependently inhibited turnover of the respective substrates, glyceraldehyde-3-phosphate and 2-phosphoglycerate (*Figure 4E*). GB-IA did not significantly inhibit GAPDH but did exhibit modest inhibitory activity towards α-enolase. GB111-NH$_2$ also showed time-dependent inhibition of GAPDH and α-enolase activity (*Figure 4F*), suggesting that it is acting as an irreversible inhibitor (*Singh et al., 2011*). Taken together, these data indicate that GB111-NH$_2$ binds covalently to reactive cysteine residues in both GAPDH and α-enolase and that binding to these cysteine residues inhibits enzyme activity.

Because GAPDH activity was recently shown to determine flux through aerobic glycolysis (*Pietzke et al., 2014*), we hypothesized that our compounds were activating NLRP3 by disrupting glycolytic flux. Productive glycolysis results in conversion of NAD+ to NADH, secretion of lactate, and ATP production (*Figure 5A*). To first test the hypothesis that GB111-NH$_2$, KA, and EB block glycolytic flux we measured the ratio of NAD+/NADH, lactate production, and intracellular ATP concentration in inhibitor-treated BMDM. LPS stimulation, which up-regulates glycolysis in macrophages (*Krawczyk et al., 2010*), resulted in an increase in NADH levels (demonstrated by a decrease in NAD+/NADH ratio) and an increase in lactate secretion (*Figure 5B–D*). GB111-NH$_2$ treatment completely blocked the lactate and NADH production induced by LPS stimulation, indicating that it directly impaired LPS-induced glycolytic flux (*Figure 5B–D*). ATP production was also significantly impaired in LPS-primed BMDM upon GB111-NH$_2$ treatment (*Figure 5E*). Notably, treatment with the NLRP3 activator nigericin did not reduce NADH levels (*Figure 5B*, *Figure 5—figure supplement 1*), indicating that the metabolic disruption that we observed with GB111-NH$_2$ is not a general feature of inflammasome activation and cell death. The GAPDH and α-enolase inhibitors KA and EB also affected metrics of glycolytic flux (*Figure 5B–D*). Finally, GB111-NH$_2$ suppressed the increase in extracellular acidification rate (ECAR) induced by glucose stimulation (*Figure 5F*). 2DG, a glycolytic inhibitor that targets hexokinase, did not dramatically impair glycolytic flux (as measured by NADH production and lactate secretion) and did not induce inflammasome formation (*Figure 5—figure supplement 2*). This indicates that severe limitation of glycolytic flux is required to activate the NLRP3 inflammasome. Furthermore, these results are in accord with recent studies showing that inhibiting GAPDH, and not enzymes in upper glycolysis, is flux-limiting in highly glycolytic cells (*Shestov et al., 2014*). Inhibitors of the TCA cycle and pentose-phosphate pathways (AA5 and 6-AN, respectively), carbohydrate metabolism pathways that are closely tied to glycolysis, also had no effect on lactate secretion or NADH production in LPS-primed BMDM and did not induce IL-1β secretion (*Figure 3A*, *Figure 5C*).

We hypothesized that, due to the dependence of macrophages on glycolytic metabolism (*Tavakoli et al., 2013*), disruption of this pathway would create a metabolic signal that is responsible for activating NLRP3. We hypothesized that supplementation of downstream metabolites of glycolysis would restore partial metabolic function and block the NLRP3-activating signal. When we cultured GB111-NH$_2$-treated BMDMs with cell-permeable versions of the terminal metabolite of glycolysis, pyruvate, or the TCA cycle metabolite succinate, we observed a dramatic reduction in the number of inflammasome foci that formed (*Figure 6A*). A structurally related metabolite, lactate, which does not fuel the TCA cycle, did not reduce GB111-NH$_2$-induced NLRP3 inflammasome formation (*Figure 6A*). Doubling the media concentration of L-glutamine, a metabolite that can be converted into succinate via anaplerosis (*Tannahill et al., 2013*), significantly reduced the number of inflammasomes that formed. Complete removal of L-glutamine from media sensitized BMDM to GB111-NH$_2$-induced inflammasome formation (*Figure 6B*). Taken together, this indicates that levels of glycolytic products that fuel downstream metabolism mediate inflammasome induction in response to glycolytic disruption.

In addition to preventing inflammasome formation, supplementation of the glycolytic product pyruvate resulted in significant reductions in caspase-1 activation, IL-1β secretion, and cell death induced by GB111-NH$_2$. Pyruvate supplementation had no effect on inflammasome signaling induced by the NLRP3 activators ATP and nigericin (*Figure 6C–E*), indicating that pyruvate does not

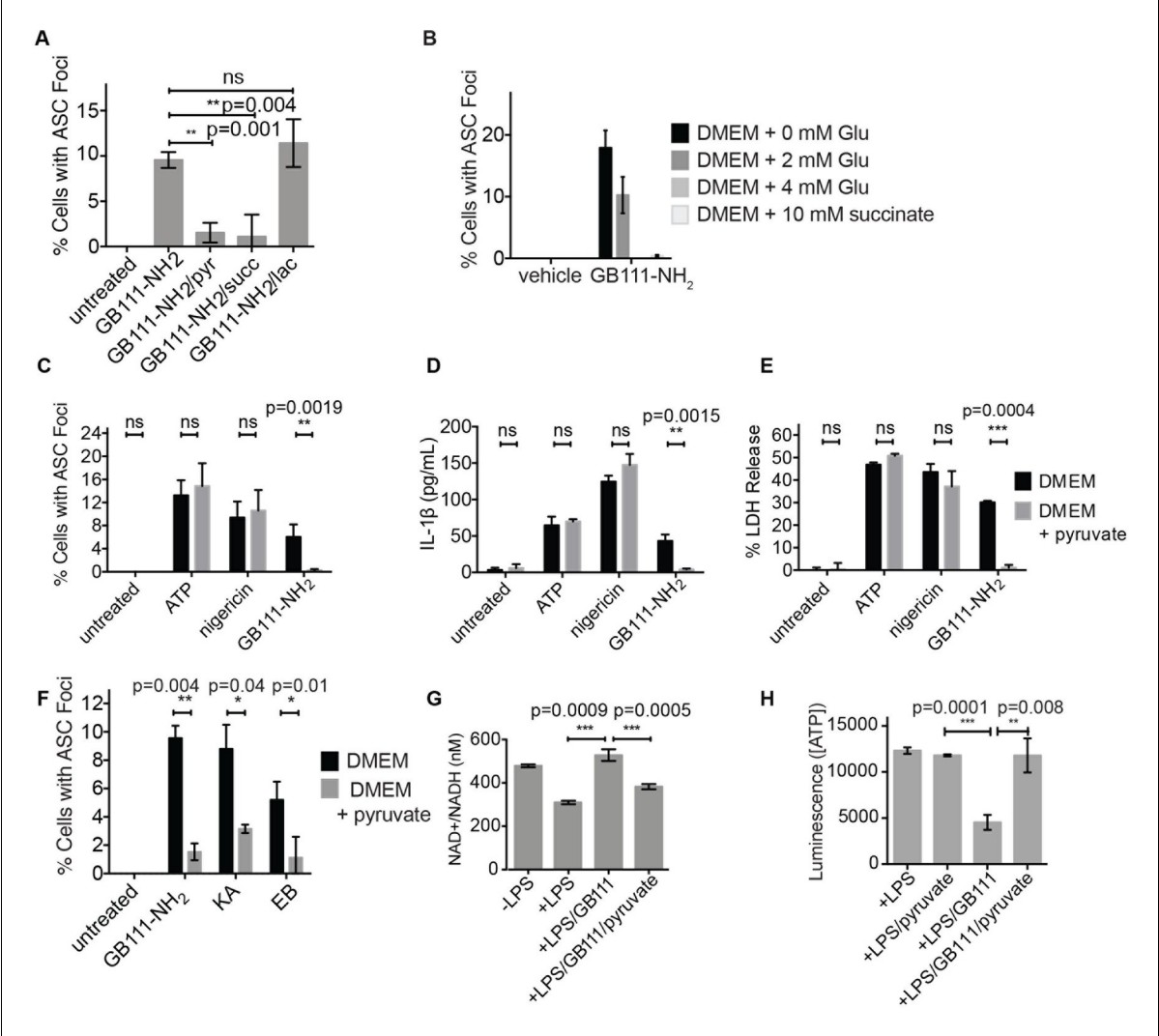

**Figure 6.** Addition of metabolites downstream of glycolysis prevents NLRP3 inflammasome activation induced by glycolytic disruption. (A) LPS-primed BMDM were treated with GB111-NH$_2$ for 2 hr in the presence of pyruvate (pyr; 1 mM) or cell-permeable esters of lactate (lac; 1 mM) and succinate (succ; 10 mM). Cells were fixed, stained for ASC and DAPI, and inflammasome foci/nuclei quantified. At least four fields of view were quantified per condition per experiment, ~2000 cells/condition. Error bars represent mean +/- sd of fields of view analyzed. (B) BMDM were primed with LPS and then treated with 10 µM GB111-NH$_2$ for 2 hr in the presence of the indicated concentrations of L-glutamine or succinate. Cells were fixed, stained for ASC and DAPI, and quantified by microscopy. Four fields of view (~2000 cells) were analyzed per condition. Error bars represent mean +/- sd of separate fields of view. (C) LPS-primed BMDM were treated with the indicated compounds in the presence or absence of pyruvate and analyzed as in (A). (D) BMDM were treated as in (C) and supernatants were analyzed for IL-1β by ELISA. (E) BMDM were treated as in (C) and cell death was measured by LDH release. (F) BMDM were treated with the indicated inhibitors, stained for ASC and DAPI, and quantified by microscopy as in (B). (G) BMDM were treated with GB111-NH$_2$ for 2 hr in the presence or absence of pyruvate (1 mM), after which cytosolic NAD+/NADH was measured. (H) BMDM were treated as in (G) and cytosolic ATP measured by ATP-coupled luminescence assay. For ELISA and LDH release data, error bars represent mean +/- sd of technical triplicate. Data were analyzed for statistical significance using an unpaired, two-tailed t test.

impair NLRP3 inflammasome signaling by a nonspecific mechanism. Pyruvate treatment also blocked inflammasome formation induced by KA and EB (*Figure 6F*) and restored NADH and ATP production in the treated cells (*Figure 6G–H*).

## NLRP3 inflammasome activation induced by GB111-NH$_2$ is mediated by NAD+/NADH imbalance and mitochondrial ROS

We hypothesized that changes in the NAD+/NADH ratio or a drop in ATP concentration could serve as a secondary signal that connects glycolytic disruption to NLRP3 inflammasome formation. To test

whether either of these signals is important, we manipulated the NAD+/NADH and ATP levels downstream of glycolysis by chemically blocking specific components of the TCA cycle and oxidative phosphorylation. We first treated LPS-primed BMDM with GB111-NH$_2$ and pyruvate to block glycolysis and stimulate downstream metabolism. We then added the succinate dehydrogenase (TCA cycle enzyme) inhibitor AA5 (*Miyadera et al., 2003*) to reduce NADH levels, the Complex I inhibitor rotenone to increase NADH levels and reduce ATP production, or the ATP synthase inhibitor Oligomycin A to only inhibit ATP synthesis by oxidative phosphorylation (*Figure 7A–B*). We found that AA5 addition partially reversed the protection conferred by pyruvate, as demonstrated by an increase in the number of inflammasome complexes. Rotenone treatment suppressed inflammasome formation more than pyruvate alone. Oligomycin A induced a small but statistically insignificant increase in the number of inflammasomes that formed (*Figure 7C*). The number of inflammasomes positively correlated with a drop in NADH production (an increase in NAD+/NADH ratio) (*Figure 7D*), while ATP concentration exhibited no correlation with the numbers of inflammasomes (*Figure 7—figure supplement 1*). Interestingly, rotenone treatment was sufficient to completely abrogate inflammasome formation induced by GB111-NH$_2$ (*Figure 7E*), conditions under which we also observed a significant decrease in the NAD+/NADH ratio (*Figure 7F*). These data suggest that the inability to produce NADH, and not ATP, is predictive of NLRP3 inflammasome formation upon glycolytic disruption by GB111-NH$_2$. It should be noted, however, that the α-enolase inhibitor EB did not induce a significant NAD+/NADH ratio defect (*Figure 5*), suggesting that either EB induces inflammasome activation through a distinct mechanism from GB111-NH$_2$ and KA, or that there are additional universal signals responsible for inflammasome activation downstream of glycolytic disruption.

Mitochondrial ROS and K+ efflux are proposed to be unifying signals preceding NLRP3 inflammasome formation (*Tschopp and Schroder, 2010*; *Muñoz-Planillo et al., 2013*). Therefore, we wanted to determine whether either of these signals is relevant to NLRP3 inflammasome activation induced by disruption of glycolysis. We stained BMDM with MitoSOX, a dye that reports accumulation of mitochondrial ROS, and observed that GB111-NH$_2$ induced an increase in cellular MitoSOX fluorescence that was abrogated by addition of pyruvate (*Figure 7G*). We also found that the ROS scavenger 4-hydroxyTEMPO (4-HT) prevented GB111-NH$_2$-induced caspase-1 cleavage and activation (*Figure 7H*). Addition of extracellular K+, in contrast, did not reduce the number of inflammasome foci in GB111-NH$_2$-treated BMDMs (*Figure 7I*) or impair GB111-NH$_2$-induced cell death (*Figure 7—figure supplement 2*), indicating that mitochondrial ROS, but not K+ efflux, is required for GB111-NH$_2$-induced NLRP3 activation and pyroptosis.

## *Salmonella typhimurium* infection induces NLRP3 inflammasome formation by disruption of host cell metabolism

We and others have shown that the intracellular pathogen *Salmonella typhimurium* (*S. typhimurium*) requires glucose and its own glycolytic enzymes for intracellular replication (*Bowden et al., 2009*; *2014*). In addition, host defense against *S. typhimurium* requires the NLRC4 and NLRP3 inflammasomes (*Broz et al., 2010*). While it is clear that *S. typhimurium* flagellin and type 3 secretion system proteins activate NLRC4 via NAIP proteins (*Zhao et al., 2011*), the mechanism by which NLRP3 is activated is not well understood. We hypothesized that *S. typhimurium* infection may stimulate NLRP3 through disruption of host cell metabolic pathways by co-opting cellular resources during intracellular replication. To test this notion, we infected naïve BMDM with *S. typhimurium* grown to stationary phase (conditions that lead to NLRP3-dependent inflammasome activation [*Broz et al., 2010*]). In this infection model, inflammasome complexes begin forming at ∼11 hr post-infection and progressively accumulate. We confirmed that infection with either wildtype *S. typhimurium* or *S. typhimurium* lacking the SPI-1 secretion system (which genetically limits *S. typhimurium* to activate NLRP3) induced inflammasome formation (*Figure 8A–B*). In addition, the percentage of cells with ASC foci was similar in magnitude to GB111-NH$_2$ and alum treatment but lower than nigericin, ATP, or log phase *S. typhimurium* stimulation (*Figure 8C*). Using this infection model, we assessed the extent to which intracellular *S. typhimurium* utilize host-cell glucose by culturing infected BMDM with the fluorescent glucose analog 2-(*N*-(7-nitrobenz-2-oxa-1,3-diazol-4-yl)amino)-2-deoxyglucose (2-NBDG). We lysed infected BMDM at 5 hr post-infection (while bacteria are still intracellular), harvested the bacterial fraction of BMDM lysates, and measured 2-NBDG fluorescence. We observed fluorescent signal that was dependent on infection in 2-NBDG-treated macrophages, indicating that the bacterial fraction took up the fluorescent glucose analog from the host cell (*Figure 8D*). We also

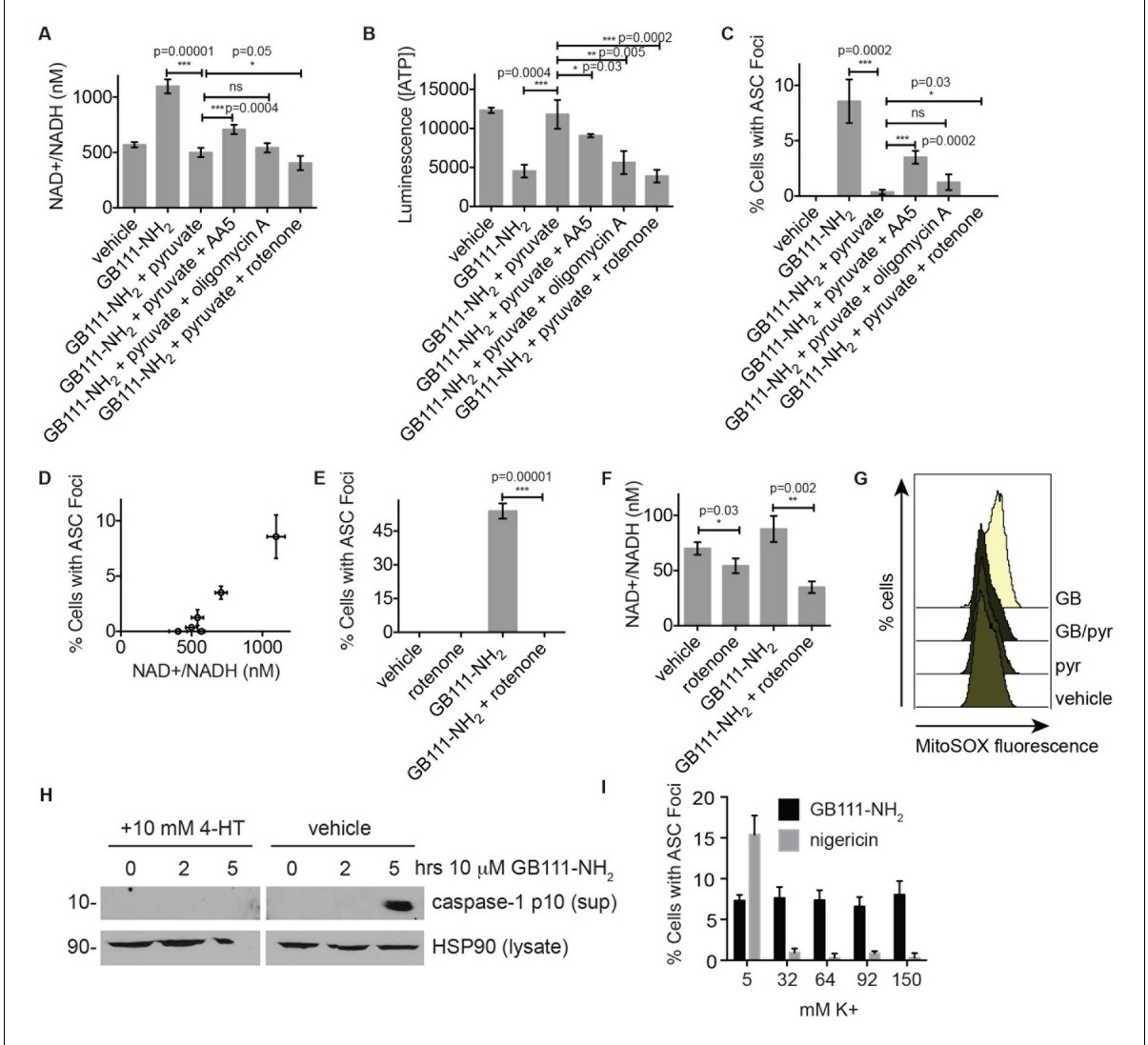

**Figure 7.** NAD+/NADH ratio elevation and mitochondrial ROS accumulation are signals for NLRP3 inflammasome formation downstream of glycolytic disruption. (A) LPS-primed BMDM were treated with the indicated compounds (GB111-NH$_2$ - 10 µM, sodium pyruvate - 1 mM, AA5 - 10 µM, oligomycin A - 1 µM, rotenone - 5 µM) for 2 hr, after which cells were fixed, stained for ASC and DAPI, and visualized by microscopy. (B) Cells were treated as in (A) and cytosolic NAD+/NADH measured. (C) Cells were treated as in (A) and cytosolic ATP measured by ATP-coupled luminescence assay. (D) % Cells with ASC foci values from (A) are plotted against NAD+/NADH values from (B). Error bars are representative of mean +/- sd of technical triplicate from (A) and (B). (E) LPS-primed BMDM were treated with vehicle or 10 µM GB111-NH$_2$ in the presence or absence of 5 µM rotenone for 2 hr. Cells were fixed, stained for ASC and DAPI, and analyzed by microscopy. Four fields of view were collected per condition (~2000 cells). (F) Cells were treated as in (E) and cytosolic NAD+/NADH analyzed. Error bars represent mean +/- sd of technical triplicate. (G) BMDM were treated with 10 µM GB111-NH$_2$ or vehicle in the presence or absence of 1 mM pyruvate (pyr) and stained with MitoSOX (2.5 µM). Cells were analyzed for MitoSOX uptake by flow cytometry. (H) LPS-primed BMDMs were treated with GB111-NH$_2$ in the presence or absence of 4-hydroxyTEMPO (4-HT). Whole cell lysates and cell supernatants (sup) were separated by SDS-PAGE and analyzed by western blot to detect the active p10 form of caspase-1. HSP90 serves as loading control. (I) BMDM were treated with nigericin (12.5 µM) or GB111-NH$_2$ (10 µM) in Ringer's buffer with increasing concentrations of K+. Cells were fixed, stained for ASC and DAPI, and inflammasome foci/nuclei quantified.

The following figure supplements are available for figure 7:

**Figure supplement 1.** ATP concentration does not correlate with inflammasome numbers.

**Figure supplement 2.** K+ efflux is not required for GB111-NH$_2$-induced pyroptotic cell death.

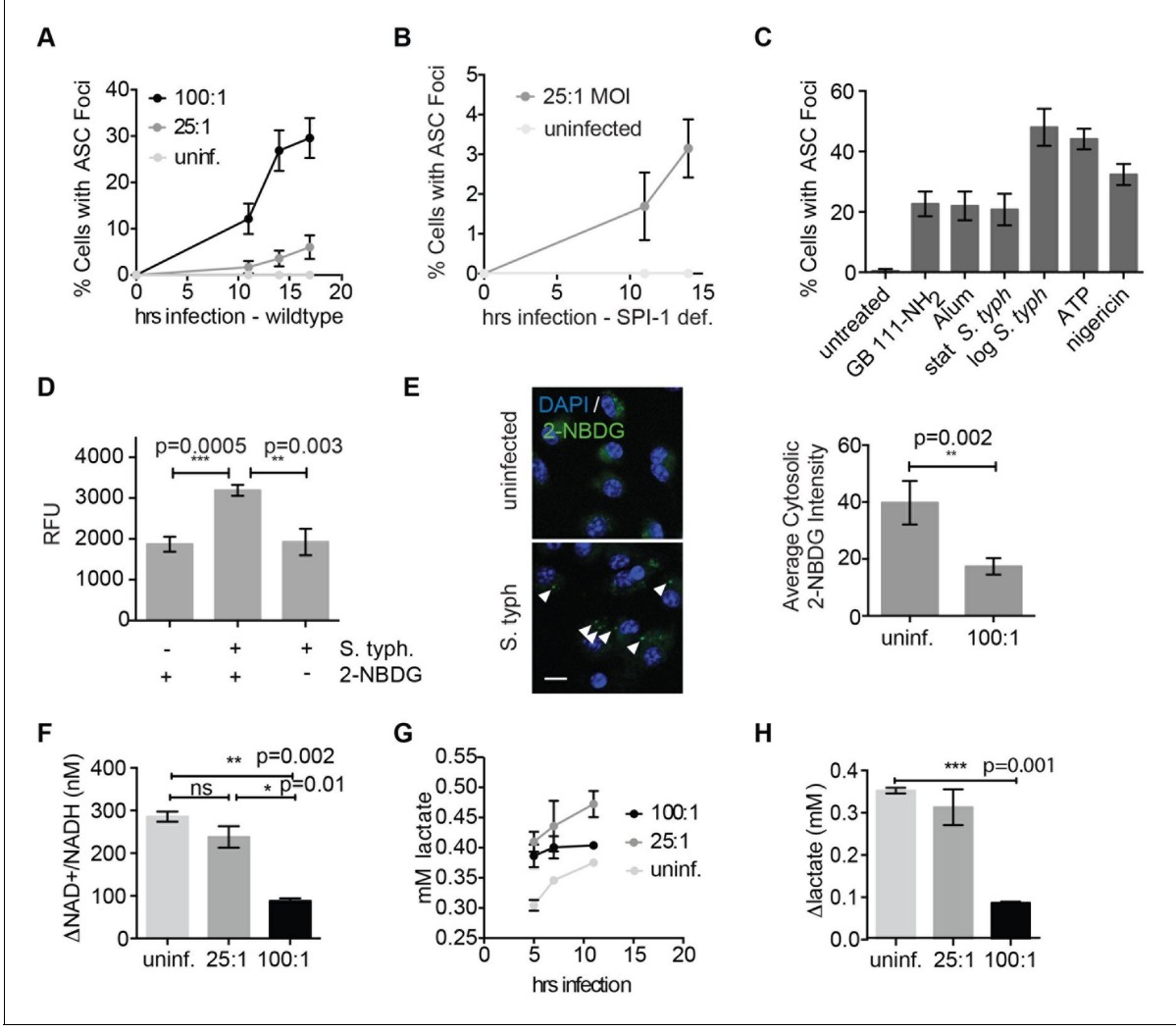

**Figure 8.** *Salmonella typhimurium* disrupts host cell glycolysis. (A) BMDMs were infected with *S. typhimurium* strain SL1344 grown to stationary phase and infected at the indicated multiplicity of infection (MOI; 25:1 and 100:1). At the indicated timepoints, cells were fixed and stained for ASC and DAPI. Inflammasome foci/nuclei were quantified. (B) BMDM were infected with △orgA (SPI-1 deficient) *S. typhimurium* grown to stationary phase. Cells were fixed at the indicated timepoints, stained for ASC and DAPI, and foci/nuclei quantified. (C) Cells were treated with the indicated compounds or infected with *S. typhimurium* grown to stationary phase (100:1 MOI) or log phase (10:1 MOI). Cells were fixed, stained for ASC, and ASC foci/nuclei were quantified. (D) BMDM were infected with 100:1 MOI stationary phase *S. typhimurium* for 5 hr. 2-NBDG (10 µM) or vehicle was added to media 2 hr post-infection. Cells were washed, lysed, intracellular bacteria sedimented from whole cell lysate via centrifugation, resuspended, and bacterial fluorescence (abs/em 465/540) analyzed by plate reader. (E) BMDM were treated as in (D), fixed, stained for DAPI, and visualized by confocal microscopy. *Left:* Representative image. White arrowheads indicate cytosolic *S. typhiurium* positive for 2-NBDG and DAPI. Scale bar 10 µM. *right:* 2-NBDG signal in areas of cytosol negative for DAPI (*S. typhimurium* negative) was measured. Cytosolic regions from ~200 discrete cells from 4 fields of view were measured per condition. Error bars represent mean +/- sd of different fields of view. (F) BMDMs were infected with stationary phase *S. typhimurium* and analyzed for cytosolic NAD+/NADH. △NAD+/NADH indicates the difference between the ratio measured at 11 hr and 5 hr post-infection. (G) BMDMs were infected with stationary phase *S. typhimurium* and levels of lactate in the supernatant analyzed at the indicated timepoints. (H) Quantification of the difference between lactate secretion measured at 11 hr and 5 hr post-infection.

fixed uninfected and infected 2-NBDG-treated BMDM and analyzed the pattern of 2-NBDG fluorescence by confocal microscopy. We observed the presence of strongly 2-NBDG fluorescent punctae that resemble *S. typhimurium* and colocalize with Hoechst stain (which stains both host cell and bacterial DNA) in infected BMDM. Furthermore, we quantified the portions of cytosol that were not Hoechst-positive and observed a significant decrease in 2-NDBG fluorescence in the cytosol of infected BMDM compared to uninfected BMDM (*Figure 8E*). These data indicate that intracellular *S. typhimurium* derive glucose from host macrophages and reduce host glucose availability.

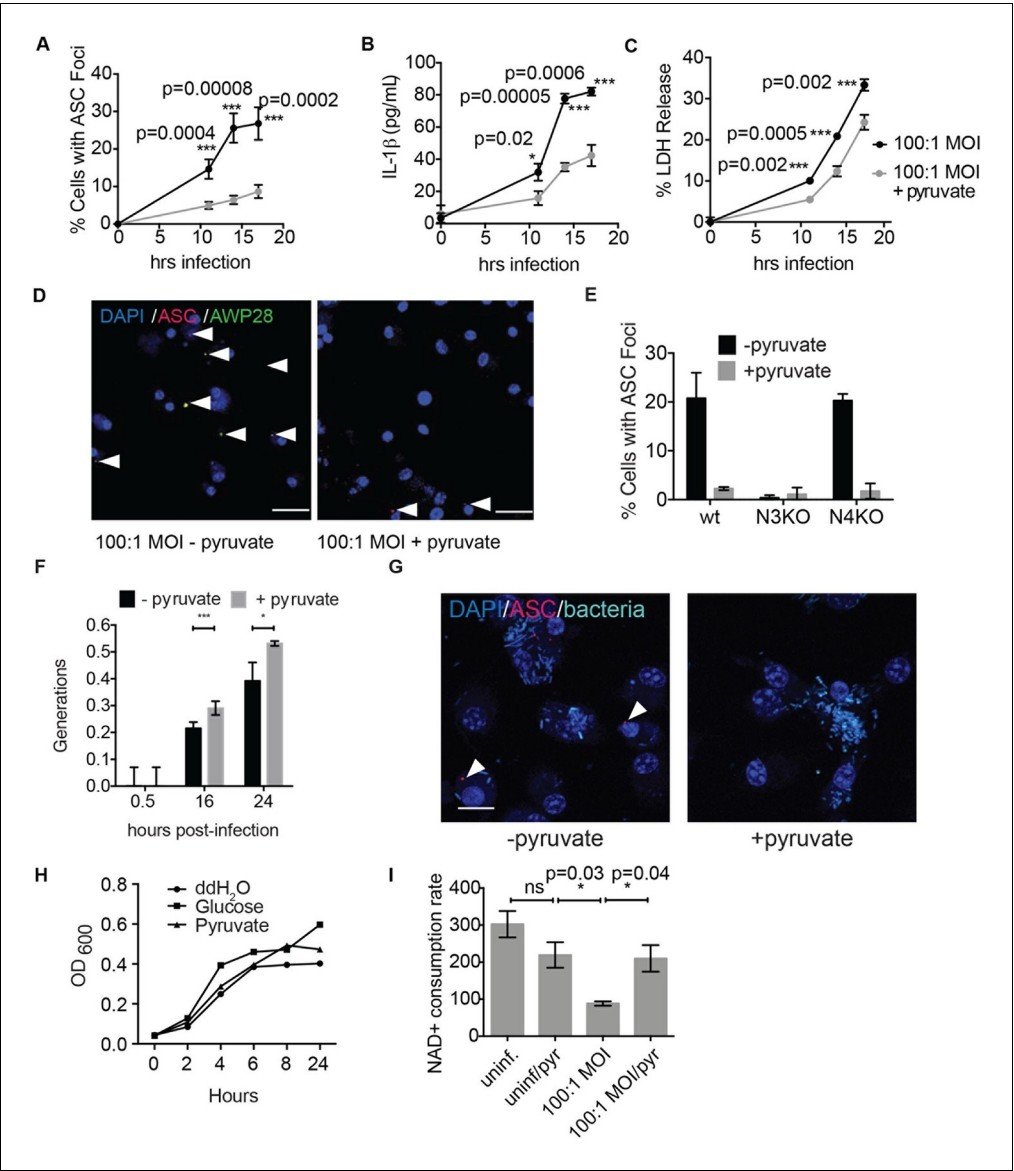

**Figure 9.** Disruption of glycolysis by *Salmonella typhimurium* activates the NLRP3 inflammasome. (**A–D**) BMDMs were infected with *S. typhimurium* grown to stationary phase in the presence or absence of 1 mM pyruvate and (**A**) cells were fixed and stained for ASC and DAPI. ASC foci/nuclei were quantified. At least four fields of view (~2000 cells) were analyzed per condition. (**B**) IL-1β secretion was analyzed by ELISA, (**C**) Cell death was measured by LDH release. (**D**) Representative image from (**A**). White arrowheads indicate inflammasome foci. Scale bar 30 μm. (**E**) BMDM of the indicated genetic backgrounds were infected with stationary phase *S. typhimurium* (100:1 MOI) in the presence or absence of pyruvate. Inflammasome foci were quantified at 17 hr post-infection. (**F**) BMDMs were infected with stationary phase *S. typhimurium* 12,023 (25:1) expressing a replication plasmid. Generations were quantified at the indicated timepoints post-infection. Data are representative of n=3 experiments. (**G**) BMDMs were infected with stationary phase *S. typhimurium* 12,023 (100:1 MOI) constitutively expressing EGFP. Cells were fixed at 17 hr post-infection, stained for ASC, and infection visualized by confocal microscopy. Scale bar 15 μm. (**H**) Minimal medium containing vehicle, 2 mM glucose, or 2 mM pyruvate was inoculated with wildtype *S. typhimurium*. Bacterial growth was measured by analyzing $OD_{600}$. (**I**) Cytosolic NAD+/NADH was analyzed at 5 and 11 hr post infection with *S. typhimurium* (100:1 MOI) or vehicle (uninf.) in the presence or absence of 1 mM pyruvate (pyr). NAD+ consumption rate indicates the difference in NAD+/NADH ratio between 5 and 11 hr post-infection. Data are representative of n=3 experiments. For LDH release, ELISA, and metabolic assays, error bars indicate mean +/- sd of technical triplicate. Data were analyzed for statistical significance using an unpaired, two-tailed t test.

*Figure 9 continued on next page*

*Figure 9 continued*

The following figure supplements are available for figure 9:

**Figure supplement 1.** Pyruvate prevents *S. typhimurium*-induced inflammasome formation in a dose-dependent manner.

**Figure supplement 2.** Inflammasome formation and cell death induced by log phase *S. typhimurium* infection are unaffected by pyruvate.

We measured glycolytic flux in infected BMDM to assess the effect of limited glucose availability on the host macrophages. Importantly, as observed for GB111-NH$_2$ treatment, we observed reduced production of NADH and lactate (*Figure 8F–H*) that correlated with the multiplicity of infection and magnitude of inflammasome formation in host cells (*Figure 8A*, *Figure 8F–H*). These metabolic defects appeared on a similar timescale as initiation of NLRP3 inflammasome formation, suggesting that infection with *S. typhimurium* has a direct effect on glycolytic flux in host cells.

Consistent with our findings using glycolytic inhibitors, we also observed that supplementation of cells with the glycolytic end product pyruvate significantly reduced inflammasome formation, IL-1β secretion, and cell death induced by *S. typhimurium* infection (*Figure 9A–D*). Pyruvate was effective at blocking inflammasome formation in BMDMs infected with both wildtype *S. typhimurium* and *S. typhimurium* defective for the SPI-1 secretion system (*Figure 9—figure supplement 1*). Notably, pyruvate did not completely block ASC focus formation, IL-1β secretion, and death induced by *S. typhimurium* infection, which could be due to compensatory activation of other inflammasomes such as the non-canonical caspase-11 inflammasome (*Broz et al., 2012*), or because *S. typhimurium* could also partially co-opt host pyruvate.

We did not observe inflammasome focus formation in *Nlrp3 -/-* BMDMs upon infection with stationary phase *S. typhimurium*. In contrast, *Nlrc4 -/-* BMDMs had a similar number of inflammasome foci upon infection as wildtype macrophages (*Figure 9E*). Pyruvate did not prevent inflammasome formation or cell death induced by infection with *S. typhimurium* in log phase growth (*Figure 9—figure supplement 2*), an infection model that activates the NLRC4 inflammasome. We also verified that pyruvate was not blocking inflammasome formation by inhibition of bacterial replication using both an intracellular replication reporter plasmid (*Helaine et al., 2010*) and by monitoring bacterial replication by microscopy (*Figure 9F–G*). Reporter plasmid expression over the course of the intracellular replication assay indicates that intracellular *S. typhimurium* are viable in BMDM cultured in DMEM with or without pyruvate (*Helaine et al., 2010*). In vitro replication assays demonstrated that *S. typhimurium* grew at a similar rate in minimal media with glucose or pyruvate as a carbon source (*Figure 9H*), further indicating that pyruvate supplementation affects host cell recognition of intracellular bacteria rather than bacterial dynamics. Importantly, we found that the NAD+ consumption rate increased upon pyruvate treatment (*Figure 9I*), indicating induction of productive metabolism downstream of glycolysis in infected BMDMs. Taken together, these data indicate that glycolytic perturbation is a mechanism by which innate immune cells sense altered homeostasis during *S. typhimurium* infection and induce a pro-inflammatory response via NLRP3 inflammasome formation and pyroptotic cell death.

## Discussion

The inflammasome is a multiprotein complex that forms in response to various pathogen- and danger-associated signals. Formation of the inflammasome leads to processing and secretion of pro-inflammatory cytokines to activate the immune system (*Lamkanfi and Dixit, 2014*; *Biswas and Mantovani, 2012*). While inflammasome formation and pyroptotic cell death are critical for fighting infection and also contribute to inflammation in diseases including type II diabetes, obesity, and atherosclerosis (*Kuemmerle-Deschner et al., 2011*; *Wen et al., 2011*; *2012*), the signals that trigger caspase-1 activation remain poorly understood. In this study, we used a small molecule, GB111-NH$_2$, to identify two glycolytic enzymes that regulate inflammasome formation. When functionally blocked, innate immune cells sense metabolic perturbation as a danger signal, resulting in inflammasome formation, caspase-1 activation, and cytokine secretion. Our results using this molecule and

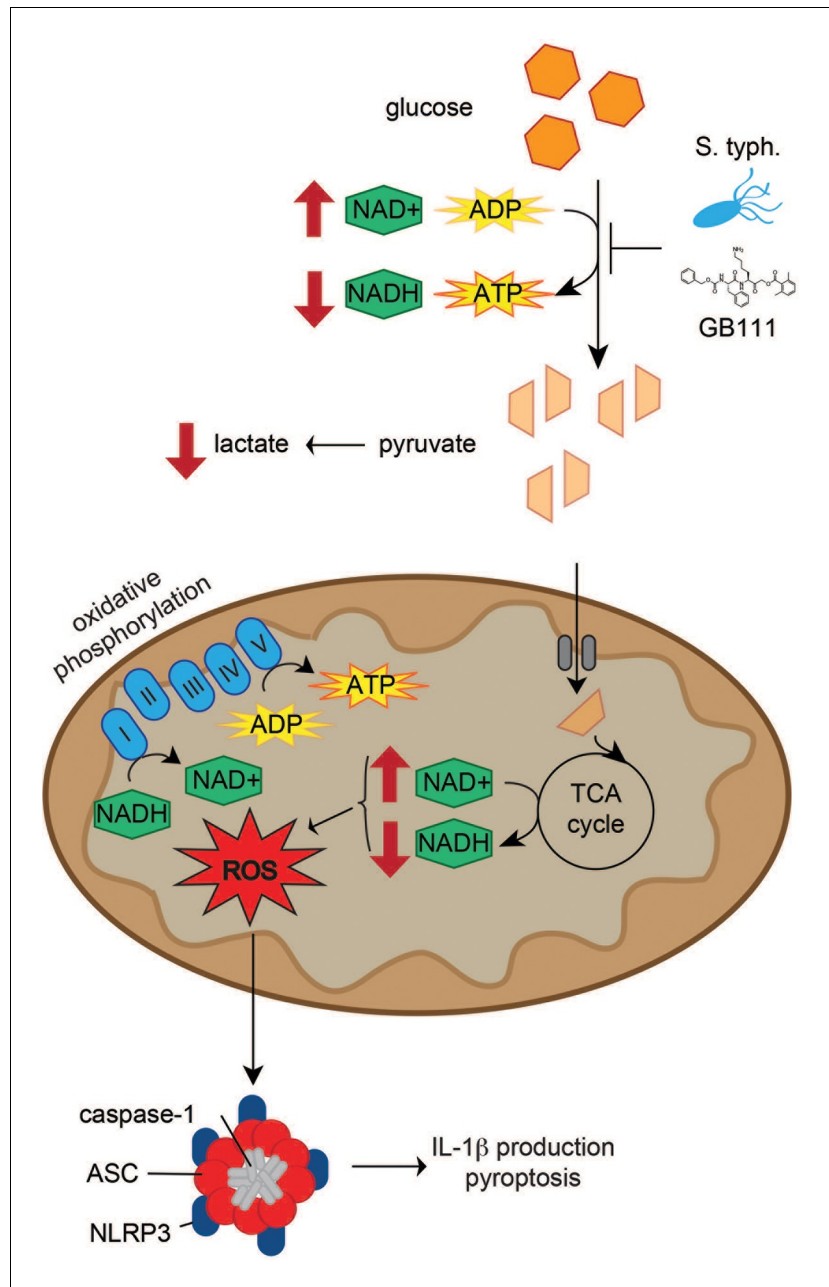

**Figure 10.** Model of NLRP3 inflammasome activation induced by disruption of glycolytic flux. *S. typhimurium* and chemical inhibitors disrupt glycolytic flux in LPS-primed BMDM, resulting in an increase in NAD+/NADH, a decrease in ATP production, and a decrease in lactate secretion. TCA cycle metabolism is also impaired, potentiating the elevated NAD+/NADH ratio into the mitochondria. Mitochondrial ROS are produced by glycolytic disruption and trigger NLRP3 inflammasome formation, IL-1β production, and pyroptosis.

other established inhibitors of these enzymes suggest that disrupting glycolytic flux serves as a trigger for inflammation and cell death in macrophages. Disturbance of glycolytic flux by the intracellular pathogen *S. typhimurium* similarly results in inflammasome formation and pyroptotic cell death in an effort to clear the pathogen. Restoration of metabolism downstream of glycolytic disruption by GB111-NH$_2$ or *S. typhimurium* infection was sufficient to abrogate the inflammasome response by restoring NADH production and preventing mitochondrial ROS production.

Though the enzymes and metabolites involved in glycolysis are well established, the specific mechanisms that limit glycolytic flux are not well understood. The irreversible reactions within

glycolysis, catalyzed by hexokinase, phosphofructokinase, and pyruvate kinase, were historically thought to be rate limiting. However, recent metabolite flux analyses have also suggested that flux through GAPDH, the enzyme separating upper and lower glycolysis, is rate-limiting under nutrient-rich conditions (*Shestov et al., 2014*). Under similar conditions, we also observed that targeting GAPDH or enzymes in lower glycolysis is sufficient to disrupt glycolytic flux and activate the NLRP3 inflammasome. Conversely, targeting entry into glycolysis using 2DG was not sufficient to disrupt glycolysis in highly glycolytic cells, as has previously been reported (*Pietzke et al., 2014*), or to activate NLRP3. This reaffirms previous observations that the pre-existing metabolic state of the cell and the point of intervention are equally important for effectively limiting glycolytic flux (*Sengupta et al., 2013*). Disruption of glycolytic flux led to an NAD+/NADH imbalance and induced mitochondrial ROS accumulation, which has been shown here and elsewhere (*Tschopp and Schroder, 2010*; *Zhou et al., 2010*) to activate the NLRP3 inflammasome (*Figure 10*).

During infection with *S. typhimurium*, inflammasome activation is an especially important mechanism of host response because, though it kills the host cell, it initiates inflammatory signals that activate the immune system and combat infection (*Storek and Monack, 2015*). Two inflammasome complexes, NLRP3 and NLRC4, are required to fully combat infection (*Broz et al., 2010*). The NLRC4 inflammasome responds to a clear pathogen-associated molecular pattern presented by *S. typhimurium*—cytosolic flagellin and type 3 secretion system components (*Zhao et al., 2011*). Here, we provide evidence that NLRP3 activation results in response to another effect of *S. typhimurium* colonization of the host cell, namely disruption of host cell glycolytic metabolism. This could explain a recent study showing that mutants of *S. typhimurium* defective for the TCA cycle enzyme aconitase induce a more rapid NLRP3-dependent immune response in vivo (*Wynosky-Dolfi et al., 2014*). We reason that aconitase deficiency would force *S. typhimurium* to rely even more heavily on glycolysis to survive within the host. These *S. typhimurium* mutants would likely disrupt cellular glycolysis more quickly and thus activate NLRP3 more rapidly. It is also interesting that, in long-term models of *S. typhimurium* infection, the bacteria preferentially resides within alternatively activated or 'M2' macrophages, which primarily utilize oxidative metabolism rather than glycolysis (*Eisele et al., 2013*). Thus, the macrophages in which *Salmonella* survives the longest are those in which host metabolic pathways are minimally perturbed, enabling prolonged infection without invoking an inflammasome response.

These findings additionally shed light on recent work highlighting the connection between metabolic changes and immune system activation (*Blatnik et al., 2008*; *Chawla et al., 2011*; *Young et al., 1984*). For example, metabolic sensing by immune cells has been recently shown to drive NLRP3-dependent IL-1β release and inflammation in diseases ranging from type II diabetes and obesity to Muckle-Wells syndrome (*Kuemmerle-Deschner et al., 2011*; *Strowig et al., 2012*), though the specific mechanisms driving macrophage and NLRP3 activation in these diseases have remained unclear. We speculate that, since glucose metabolism is often impaired in these diseases, glycolytic impairment may be the mechanism driving NLRP3-dependent inflammation. Restoring glycolysis or downstream TCA cycle metabolism through supplementation with specific metabolites or activation of glycolytic enzymes could be therapeutically useful for dampening inflammation and associated immunopathology.

In summary, our results suggest that inhibition of glycolysis creates a unique metabolic state that activates the NLRP3 inflammasome. They suggest that innate immune cells sense perturbed metabolite production and flux through the glycolytic pathway, in turn activating NLRP3 to initiate inflammatory responses. Inhibitors of flux-limiting enzymes and *S. typhimurium* effectively limited glycolysis through distinct mechanisms, each resulting in NLRP3-mediated inflammasome formation and pyroptosis. Glycolytic disruption may be a broadly relevant mechanism of NLRP3 activation triggered in response to metabolic parasitism by microbes. Moreover, this pathway may also provide novel avenues for treating diseases in which NLRP3-driven inflammation results in pathology.

## Materials and methods

### Compound information

See below for synthesis and characterization of GB111-NH$_2$ and analogs. NMR spectra were recorded on a Varian 400 MHz (400/100) or a Varian Inova 500 MHz (500/126 MHz) equipped with a

pulsed field gradient accessory. Chemical shifts (∂) are reported in parts per million (ppm) downfield from tetramethylsilane and are reference to the residual protium signal in the NMR solvents. Data are reported as follows: chemical shift, multiplicity (s=singlet, d=doublet, t=triplet, m=multiplet and q=quartet), coupling constant (J) in Hertz (Hz) and integration. E64d (Enzo Lifesciences, Farmingdale NY), leupeptin (Sigma Aldrich, St. Louis MO), CA074Me (EMD Millipore, Hayward CA), LPS from *E. coli* 0111:B4 (Sigma Aldrich), 6-aminonicotinamide (Santa Cruz Biotech, Santa Cruz CA), Imject Alum (Pierce Biotechnology, Rockford IL), Atpenin A5 (Santa Cruz Biotech), N-ethylmaleimide (Sigma), rotenone (Sigma), oligomycin A (Cayman Chemical, Ann Arbor MI), nigericin (Cayman Chemical), MitoSOX (Life Technologies, Carlsbad CA), ATP (Sigma), and koningic acid (Adipogen, Switzerland) were purchased from commercial sources, dissolved in vendor-recommended solvents, and used without further purification. ENOblock (*Jung et al., 2013*) was a generous gift from Dr. Darren Williams.

## Bacterial strains
Strains used in this study were *Salmonella typhimurium* NCTC 12023 and ATCC SL1344. Bacteria were grown in LB at 37°C with aeration and supplemented with 0.2% arabinose if needed to induce expression of fluorescent proteins.

## Mice
Mice lacking *Pycard, Nlrp3, Nlrc4,* and *Aim2* have been previously described (*Jones et al., 2010*; *Broz et al., 2010*; *Kayagaki et al., 2011*; *Mariathasan et al., 2004*). Mice were maintained following guidelines approved by the Stanford University School of Medicine Administrative Panel on Laboratory Animal Care.

## Cell culture protocols
BMDM were isolated by culturing mouse bone marrow in DMEM with 2 mM L-glutamine, 10% FBS, and 10 ng/mL recombinant mouse M-CSF (eBioscience, San Diego CA) for 5 days in petri dishes. After 5 days, the cell monolayer was washed several times with sterile PBS to remove cell debris and then the BMDM harvested using CellStripper (Corning CellGro, Manassas VA). BMDM were then plated for experiments, frozen, or cultured for up to a week. One day prior to treatment, cells were seeded in 6 well plates at a density of $1\text{-}2\times10^6$ cells/well (or $2\times10^5$ cells/well of 24-well dish, or $3\times10^4$ cells/well of 96-well dish). C57BL/6 SV40-immortalized macrophages were cultured in RPMI with 10% FBS and 2 mM L-glutamine and were a generous gift from Petr Broz.

## Replicates and statistical analyses
In this study, biological replicates indicate replicates of the same experiment conducted upon separately seeded cultures on separate days. Technical replicates indicate separate measurements made on cells seeded on the same day and treated simultaneously. The number of biological replicates is indicated in the figure legends and was generally n=3. For plate reader-based assays, experiments were generally conducted in technical triplicate as recommended by assay manufacturers. For microscopy experiments, at least four fields of view were generally analyzed – covering the four quadrants of the cover slip. Within each quadrant, a field was chosen at random using the DAPI channel (to simply find a region that contained cells). Each field of view was counted as a technical replicate because it was a separate measurement of a singly seeded culture. When ascertaining whether differences between samples were statistically significant, an unpaired, two-tailed t test was used. This makes the assumptions that the two samples under analysis were approximately normally distributed and had equal variances. $p<0.05$ was considered significant. Because measurements were taken within linear range of the detection method (i.e., below saturation and above noise for absorbance-based plate reader assay, within linear range of detector for flow cytometry measurements), etc, technical replicates should be normally distributed around the mean.

## LPS priming and inflammasome activation
BMDM were primed with 100 ng/mL LPS-EK (Invivogen, San Diego CA) or LPS 0111:B4 (Sigma) in DMEM for 3 hr before addition of inflammasome activating agents. GB111-NH$_2$ was added to LPS-primed BMDM at 10 µM (unless otherwise indicated) for inflammasome activation. The canonical

NLRP3 activators ATP and nigericin were added to LPS-primed BMDM at 5 mM and 12.5 µM, respectively, typically for 1 hr. Alum (Pierce) was used at a concentration of 100 µg/mL for 5 hr.

### *Salmonella typhimurium* infections

For stationary phase infections, *S. typhimurium* grown to stationary phase (typically overnight culture in LB) were centrifuged onto BMDM for 10min at 500g. After 1 hr, medium was switched to DMEM with 100 µg/mL gentamicin sulfate (Sigma) to kill extracellular bacteria. After 1 hr, cells were washed with plain DMEM and then incubated in DMEM with 10 µg/mL gentamicin sulfate for the remainder of the infection. For log phase infections, *S. typhimurium* in log phase growth (typically a 4 hr subculture of a 1:50 dilution of an overnight culture) were centrifuged onto BMDM for 10min at 500g in antibiotic-free DMEM. Unless otherwise stated, samples were analyzed after 1 hr of log phase infection.

### Probe labeling

Probes were diluted to the desired final concentration (1 µM for AWP28, 250 nM for BMV109) from a 1000x stock solution in DMSO directly in the media of the cell monolayer. Cells were labeled for the final hour of treatment at 37°C prior to sample preparation and analysis. For gel labeling experiments, labeled cell monolayers were washed in PBS and lysed directly with 50 µL sample buffer. For harvested supernatants, following treatment the supernatant was removed and proteins precipitated by adding 4 equivalents of cold acetone. Samples were incubated in acetone overnight at -20°C, then proteins pelleted by centrifugation for 5 min at 2000 rpm. Acetone was aspirated and protein pellets dried for 30 min at 37°C before addition of sample buffer. Samples were resolved by SDS-PAGE and visualized on a Typhoon flatbed fluorescent laser scanner (GE Healthcare, United Kingdom).

### Western blots

Following separation of samples by SDS-PAGE and transfer to 0.2 µM nitrocellulose resin (BioRad, Hercules CA), the following antibodies were used. For cell lysates: anti-caspase-1 p10 (1:200, Santa Cruz Biotechnology #514), anti-HSP-90 (1:1000, BD Biosciences, San Jose CA), anti-NLRP3 (1:500, R&D Systems, Minneapolis MN), anti-α-tubulin (1:10000, Sigma), anti-GAPDH (1:1000, Santa Cruz Biotechnology C-9), anti-α-enolase (1:1000, Cell Signaling Technology, Danvers MA). For cell supernatant: anti-IL-1β (1:200, Cell Signaling Technology). HRP-conjugated α-mouse and α-rabbit secondary antibodies were from GE Healthcare.

### ELISA protocols

BMDMs were seeded in triplicate in 96-well plates at a density of $3x10^4$ cells/well. Following treatment, the supernatant was removed and IL-1β, IL-6, or TNF-α release was measured using a Mouse IL-1β, IL-6, or TNF-α READY-SET-GO ELISA kits (eBioscience) according to the manufacturer's instructions.

### LDH release assays

BMDMs were seeds in triplicate in 96-well plates at a density of $3x10^4$ cells/well. Following treatment, the supernatant was removed, and the cells were lysed with 2% Triton-X-100 in D-PBS. The lysate was diluted in culture media to the original volume. LDH release was calculated as supernatant LDH activity/total LDH activity using the CytoTox 96 Assay (Promega, Madison WI).

### Microscopy and image analysis

BMDMs were seeded on poly-L-lysine coated glass coverslips in 24 well plates at a density of $2x10^5$ cells/well. Following treatment and labeling with AWP28 (1 µM for final hour of treatment), the cell monolayer was rinsed 3x with warm D-PBS and then fixed with 4% paraformaldehyde in PBS for 15 min at 37°C. The cells were washed with PBS and incubated with anti-ASC (1:200, Santa Cruz Biotechnology N-15) primary antibody in blocking buffer (3% BSA, 0.1% saponin, 0.02% sodium azide in PBS) for 30 min. The cells were washed 3x with PBS and incubated with Alexa 647 or Alexa 594-conjugated secondary antibody (both 1:1000, Invitrogen, Carlsbad CA) for 30min. The cells were washed with D-PBS, mounted in Vectashield with DAPI (Vector Labs, Burlingame CA), and imaged

on a Zeiss LSM700 confocal microscope. Snapshots of fields were taken at random (at least 4 fields/ condition using a 10x or 20x air objective, typically ~2000 cells/condition). Nuclei were counted using the ITCN plug-in in ImageJ and inflammasome (ASC and/or AWP28 positive) foci were counted using the 'Analyze Particles' function in ImageJ after automated thresholding. Replicates indicate cells plated and treated on separate days. For Annexin V and propidium iodide staining, AWP28 labeled cells on coverslips were washed with Annexin V binding buffer (10 mM HEPES pH 7.4, 150 mM NaCl, 2.5 mM CaCl$_2$) and then incubated with 1 µg/mL propidium iodide (ImmunoChemistry, Bloomington MN) and 1:50 Alexa 647 conjugated Annexin V (Invitrogen) in Annexin V-binding buffer on ice for 30 min. Cells were washed with Annexin V-binding buffer and mounted in Vectashield (Vector Labs) for immediate imaging.

## Mass- spectrometry sample preparation and analysis

### Competition proteomics and sample preparation

C57BL/6 BMDMs were seeded onto 15 cm dishes ($2x10^7$ cells/dish). The number of dishes per condition was calculated such that approximately 3 milligrams of protein were yielded per condition. The competition experiment took place as follows: For condition 1, BMDMs were incubated with 100 ng/mL LPS for 3 hr, after which 50 µM az-GB from 100x DMSO stock was added to culture media for 2 hr. For condition 2, BMDMs were incubated with 100 ng/mL LPS for 2 hr. 10 µM GB111-NH$_2$ from 1000x DMSO stock was added to the culture media for 1 hr, after which 50 µM az-GB from 100x DMSO stock was added to culture media for 2 hr. For condition 3, BMDMs were incubated with 100 ng/mL LPS for 2 hr. 50 µM GB-IA from 1000x DMSO stock was added to the culture media for 1 hr, after which 50 µM az-GB from 100x DMSO stock was added to the culture media for 2 hr. For condition 4, BMDMs were incubated with 100 ng/mL LPS for 3 hr, after which vehicle was added for 2 hr. After treatment, all cells were lifted from tissue culture dishes using CellStripper (Corning Cellgro), pelleted at 1000 rpm for 5min, washed once with PBS, and lysed on ice in D-PBS containing 1% NP-40 and 0.1% SDS. Cellular debris was pelleted by centrifugation at 14000 rpm for 15 min at 4°C. The supernatant was removed and protein concentration determined by BCA Assay (Pierce). Protein concentrations were then normalized to 2 mg/mL in PBS with 1% SDS.

### Click chemistry and streptavidin enrichment of probe-labeled proteins

Protein samples (>3 mg/condition) then underwent click chemistry. Biotin azide was added to 10 µM final concentration, fresh TCEP (Sigma) to 1 mM, TBTA (Sigma) to 100 uM, and CuSO$_4$ to 1 mM. The samples were allowed to react at room temperature for 3 hr. Proteins were then precipitated using 5 volumes -20°C acetone. After 2 hr, protein precipitates were pelleted. The pellets were washed 4x with -20°C acetone, air dried, and resuspended in PBS with 1.2% SDS. These solutions were incubated with 100 µL streptavidin-agarose beads (Thermo Scientific) at 4°C for 16 hrs. The solutions were then incubated at room temperature for 2.5 hr. The beads were washed with 0.2% SDS/ PBS (5 mL), PBS (3 x 5 mL), and water (3 x 5 mL). The beads were pelleted by centrifugation (1400 x g, 3 min) between washes.

### On-bead trypsin digestion

The washed beads were suspended in 6 M urea/PBS (500 µL) and 10 mM dithiothreitol (DTT) (from 20X stock in water) and placed in a 65°C heat block for 15 min. Iodoacetamide (20 mM, from 50X stock in water) was then added and the samples were placed in the dark and allowed to react at room temperature for 30 min. Following reduction and alkylation, the beads were pelleted by centrifugation (1400 x g, 3 min) and resuspended in 200 µL of 2 M urea/PBS, 1 mM CaCl2 (100X stock in water), and trypsin (2 µg). The digestion was allowed to proceed overnight at 37°C. The peptide digests were separated from the beads using a Micro Bio-Spin column (BioRad). The beads were washed with water (2 x 50 µL) and the washes were combined with the eluted peptides. Formic acid (15 µL) was added to the samples. These tryptic digests were stored at -20°C until mass spectrometry analysis.

### Liquid chromatography-mass spectrometry (LC-MS) analysis

LC-MS analysis was performed on an LTQ Orbitrap Discovery mass spectrometer (ThermoFisher, Waltham MA) coupled to an Agilent 1200 series HPLC. Digests were pressure loaded

onto a 250 µm fused silica desalting column packed with 4 cm of Aqua C18 reverse phase resin (Phenomenex, Torrance CA). The peptides were eluted onto a biphasic column (100 µm fused silica with a 5 µm tip, packed with 10 cm C18 and 3 cm Partisphere strong cation exchange resin (SCX, Whatman, United Kingdom) using a gradient 5–100% Buffer B in Buffer A (Buffer A: 95% water, 5% acetonitrile, 0.1% formic acid; Buffer B: 20% water, 80% acetonitrile, 0.1% formic acid). The peptides were eluted from the SCX onto the C18 resin and into the mass spectrometer following the four salt steps outlined in *Weerapana et al., 2007*. The flow rate through the column was set to ~0.25 µL/min and the spray voltage was set to 2.75 kV. One full MS scan (400–1800 MW) was followed by 8 data dependent scans of the nth most intense ions with dynamic exclusion enabled.

## Mass spectrometry data analysis

The generated tandem MS data was searched using the SEQEST algorithm against the human UNI-PROT database. A static modification of +57 on Cys was specified to account for iodoacetamide alkylation. The SEQUEST output files generated from the digests were filtered using DTASelect 2.0 to generate a list of protein hits with a peptide false-discovery rate of <5%.

When comparing results from Conditions 1–4, spectral counts were first normalized based on the spectral counts of the four endogenously biotinylated mammalian proteins, pyruvate carboxylase, 3-methylcrotonyl CoA carboxylase, propionyl CoA carboxylase, and acetyl CoA carboxylase (*Chandler and Ballard, 1985*). Condition 4 determined 'background' levels of reactivity with alkyne-biotin. Candidate proteins were those with >30 spectral counts in condition 1, >80% competition by GB111-NH$_2$ for az-GB binding in condition 2, and less than 50% competition by GB-IA for az-GB binding in condition 3. Pearson correlation between enrichment in different samples and expected enrichment was calculated for confidence in hit proteins.

## Enzyme labeling assays

Recombinant GAPDH (ScienCell, Carlsbad CA), α-enolase (BioVision, Milpitas CA) were diluted into assay buffer (50 mM Tris-HCl pH 7.4, 1.5 mM MgCl$_2$) and incubated with inhibitor or vehicle for 30 min at 37°C. After this, az-GB (50 µM) was added for 2 hr at 37°C. TAMRA-alkyne was then added under previously described Click reaction conditions (*Child et al., 2013*) to visualize az-GB-labeled protein. Reaction mixtures were separated by SDS-PAGE and visualized on Typhoon scanner.

## Enzyme activity assays

### GAPDH activity assay

Recombinant GAPDH (0.02 units) was incubated in GAPDH Assay Buffer (ScienCell) for 30 min at 37°C in the presence of inhibitor or vehicle. This mixture was then added to Assay buffer, which contains 6.7 mM phosphoglyceric acid, 3.3 mM L-cysteine, 117 µM β-NADH, 1.13 mM ATP, and 0.05 U 3-phosphoglycerate kinase in 150 µL. $A_{340}$, representing conversion of β-NADH to NAD+, was measured every minute for 30 min by plate reader (SpectraMax M5, Molecular Devices, Sunnyvale CA). Percentage inhibition was calculated as: (treatment $\triangle A_{340}$/vehicle $\triangle A_{340}$)x100.

### α-enolase activity assay

Approximately 0.013 units of recombinant α-enolase (MyBioSource.com) were incubated in assay buffer (50 mM Tris-HCl pH 7.4, 1.5 mM MgCl$_2$) for 30 min at 37°C in the presence of inhibitor or vehicle. Phosphoenolpyruvate (Sigma) was added to a final concentration of 1.5 mM. $A_{240}$, representing conversion of phosphoenolpyruvate to 2-phosphoglycerate, was measured every minute for 30 min by plate reader. Percentage inhibition was calculated as: (treatment $\triangle A_{240}$/vehicle $\triangle A_{240}$)x100.

## Metabolic assays

### NAD+/NADH assay

BMDM were plated in 96-well dishes at 50 k cells/well. The next day, cells were treated with chemical compound or infected with *Salmonella typhimurium*. Plates were centrifuged at 500g for 5 min at room temperature, after which culture medium was aspirated and 100 µL lysis buffer (Cayman Chemical) added to each well. Plates were nutated at room temperature for 30 min and then centrifuged at 1000 g for 10 min at 4°C. Supernatants were transferred to wells of a new plate, and

100 µL NAD+/NADH reaction solution (Cayman Chemical) was added to each well. After 1.5 hr, $A_{450}$ was measured.

## NADH assay
Cells were treated and lysates harvested as for the NAD+/NADH assay. After this, NAD+ was decomposed by heating at 60°C for 30 min. Then, reaction solution was added and after 1.5 hr, $A_{450}$ was measured.

## Lactate release assay
BMDM were plated in 96-well dishes at 50 k cells/well. The next day, cells were treated with chemical compound or infected with *Salmonella typhimurium* in phenol red-free DMEM. Plates were centrifuged at 500 g for 5 min at room temperature, after which 50 µL of supernatant/well was transferred to a new 96-well dish. Lactate reaction solution (50 µL; Eton Biosciences, San Diego CA) was added. After 30 min, the reaction was quenched with 50 µL/well of 0.5M acetic acid and $A_{490}$ was measured.

## ATP assay:
BMDM were plated in opaque-walled 96-well dishes at 50 k cells/well. The next day, the cells were treated with chemical compounds in 100 µL well volume. After 1 hr of treatment at 37°C, the plate was brought to room temperature for 30 min as per manufacturer's instructions (Promega – CellTiter Glo). ATP reaction mixture was added directly to wells (100 µL/well) and plate was nutated for 2 min to lyse cells. Plate was allowed to stabilize for 10-15 min at room temperature, after which luminescence was read by plate reader (1 s integration time/well).

## Seahorse analyzer assay
For ECAR measurements, BMDM were analyzed using a Seahorse XF96 Analyzer. On the day prior to the assay $8 \times 10^3$ BMDM were plated per well of a 96-well Seahorse Analyzer plate. The next day, cells were washed with and then immersed in 180 mL Assay Medium (RPMI at pH 7.4 with 2 mM L-glutamine and without HEPES or sodium bicarbonate). Cells were incubated in a $CO_2$-free incubator for 1 hr at 37°C. At initiation of assay, the plate was loaded into the Seahorse Analyzer, allowed to equilibrate, and compounds injected in Assay Medium with fresh glucose. ECAR was measured for 2 hr after compound injection. Cells were stained with Hoechst and counted after conclusion of assay. Measured ECAR values were normalized to cell number and averaged across each condition.

## Fluorescent glucose assays
The fluorescent glucose analog 2-NBDG (Cayman Chemical; Abs/Em 465/540 nm) was used to monitor glucose uptake by both infected and uninfected BMDM. BMDM were infected with SL1344 *Salmonella typhimurium* grown to stationary phase at an MOI of 100:1. After 1 hr, media was changed to DMEM with high gentamicin (100 µg/mL) to kill extracellular bacteria. After 1 hr, BMDM were washed with plain DMEM and then incubated in DMEM with low gentamicin and 10 µM 2-NBDG. For microscopy analysis, cells were fixed and mounted in Vectashield with DAPI after 4 hr of infection. 2-NDBG was imaged using 'FITC' absorption/emission settings in ZenBlack software on a Zeiss LSM700 microscope. Quantification of average cytosolic 2-NBDG fluorescence was done using ImageJ software. In the uninfected condition, cytosol was identified as 2-NBDG (+) areas proximal to nuclei. In the infected condition, cytosol was identified as areas proximal to nuclei that were not *Salmonella typhimurium* (+). 4 fields per sample were quantified and the average and standard deviation of average cytosolic 2-NDBG fluorescence measurements reported. For measurement of 2-NBDG uptake into *S. typhimurium*, after 7 hr of infection BMDM were lysed in 0.1% Triton-X-100 in PBS for 10 min. Lysates were centrifuged at 5000 g/10 min/4°C. Supernatant was aspirated and the resulting bacterial pellet resuspended in PBS, transferred to an opaque 96-well plate, and measured in triplicate on a plate reader at Abs/Em 465/540 nm.

## *Salmonella* replication assays
*Salmonella typhimurium* (strain 12023) expressing a replication plasmid were grown overnight in LB containing 0.2% arabinose. BMDM were plated in 12-well dishes at 500 k cells/well and infected

with *Salmonella typhimurium* strain NCTC 12,023 at MOI 25:1. At 12, 16, and 24 hr post-infection, BMDM were lysed and bacterial samples analyzed by flow cytometry. Generations of bacteria were calculated as previously described by Helaine et al. For in vitro growth curves, *S. typhimurium* were grown in MgM-MES minimal media supplemented with 2 mM glucose, 2 mM pyruvate, or vehicle (ddH$_2$O). OD$_{600}$ was measured at various timepoints after inoculation of culture.

## Mitochondrial ROS measurement

BMDM were plate in 12-well dishes, primed for 3 hr with 100 ng/mL LPS, and then stimulated in the presence or absence of pyruvate. BMDM were labeled for the last 15min of treatment with 2.5 μM MitoSOX Red (Life Technologies), collected, centrifuged for 5min at 2000 rpm at 4°C, then resuspending in ice cold PBS with 0.5% BSA and analyzed by flow cytometry (488 nm excitation, PE channel collection for MitoSOX Red). >25,000 cells were analyzed per condition.

## K+ efflux experiments

LPS-primed BMDM were treated with NLRP3-activating compound in Ringer's buffer with varying concentrations of K+. Osmolarity was kept constant by varying NaCl concentration accordingly.

## Synthetic protocols

**General synthesis schema.** Reagents and conditions: i. IBCF, NMM, THF, -77°C, 1 hr, then CH2N2, -77°C, 1 hr, then warm to RT, 3 hr, then 1:1 HCl:AcOH. ii. 2,3,5,6-tetrafluorophenol, KF, DMF, 80°C, 2 hr. iii. 50% TFA in DCM, 30 min. iv. 2,6-dimethylbenzoic acid, KF, DMF, 9 hr. v. Acetyl chloride, TEA, DCM, 30 min. vi. imidazole-1-sulfonyl azide, K2CO3, CuSO4, MeOH, o/n.

General procedure for chloromethylketone (CMK) synthesis

Peptide carboxylic acid (1eq), was stirred with isobutyl chloroformate (1.1 eq) and N-methyl morpholine (1.2 eq) in anhydrous THF in a bath of dry ice/isopropanol for 1 hr, after which a solution of CH$_2$N$_2$ (approximately 1.7 eq, freshly generated from diazald) was added. The mixture was stirred in dry ice/isopropanol for 1 hr, and then brought to room temperature and stirred for 3 hr. The reaction was quenched with 1:1 concentrated HCl:HOAc (v:v). Ethyl acetate was added to the crude reaction mixture and the organic layer was washed with H$_2$O, saturated NaHCO$_3$, and brine. The organic layers were pooled and dried with MgSO$_4$, and concentrated *in vacuo* to yield crude chloromethylketone.

## General procedure for acyloxymethylketone (AOMK) synthesis

Chloromethylketone (1 eq) was stirred with potassium fluoride (3 eq) in anhydrous DMF for 15 min. 2,6-dimethylbenzoic acid (1.1 eq) was added and the reaction mixture stirred for 9 hr at room temperature.

**Chemical structure 1.** Carboxybenzyl-Phe-Lys-NH$_2$ (NR-GB11).

## Synthesis of NR-GB111 (3)

Rink resin (1g, 0.59 mmol) was taken up in DMF and deprotected in 20% piperidine in DMF for 45 min at room temperature. The resin was washed with DMF. Fmoc-Lys(Boc)-OH (829 mg, 3 eq, 1.77 mmol), HOBt (239 mg, 3 eq, 1.77 mmol), and DIC (277 µL, 3 eq, 1.77 mmol) were added and the reaction mixture nutated for four hours. The resin was washed with DCM and DMF and the Fmoc group removed by incubation with 20% piperidine in DMF for 45 min. The resin was washed with DMF and Z-Phe-OH (530 mg, 3 eq, 1.77 mmol), HOBt (239 mg, 3eq, 1.77 mmol), and DIC (277 µL, 3 eq, 1.77 mmol) were added and the reaction mixture nutated overnight at room temperature. The resin was washed with DCM and DMF. The product NR-GB111 was cleaved from the Rink resin using 95% TFA, 2.5% triisopropylsilane, and 2.5% H$_2$O for 30 min. The crude was purified by HPLC (reverse phase C$_{18}$ column, CH$_3$CN/H$_2$O 0.1% TFA, 5:95 to 80:20 over 9 column volumes (CVs) Pure fractions were lyophilized and 5.55 mg (0.013 mmol, 2.2% yield) **NR-GB111 (3)** were afforded as a white powder.

[1]H NMR (500 MHz, CD$_3$OD) δ 7.36 – 7.19 (m, 10H), 5.03 (q, $J$ = 12.6 Hz, 2H), 4.38 – 4.27 (m, 2H), 3.08 (dd, $J$ = 13.7, 6.5 Hz, 1H), 2.92 (dd, $J$ = 13.7, 8.6 Hz, 1H), 2.86 (t, $J$ = 7.6 Hz, 2H), 1.93 – 1.79 (m, 1H), 1.69 – 1.53 (m, 3H), 1.47 – 1.32 (m, 2H).

HRMS (ES+): [M+H+]$^+$ calculated for C$_{23}$H$_{30}$N$_4$O$_4$ expected mass 427.2345 found 427.2345. LCMS (ES+): retention time 5.57 min.

## Synthesis of GB-IA (4)

**Chemical structure 2.** pent-4-ynamido-Phe-Lys(Boc)-OH.

## pent-4-ynamido-Phe-Lys(Boc)-OH (9)

Chlorotrityl resin (900 mg, 1.134 mmol, 1 eq) was swelled in anhydrous DCM. Fmoc-Lys(Boc)-OH (798 mg, 1.701 mmol, 1.5 eq) and DIPEA (402 µL, 2.31 mmol, 2 eq) were added and the reaction mixture nutated for 3 hr at room temperature. 500 µL anhydrous methanol was added for 30 min. The resin was washed with DCM, DMF, and then resin loading measured (0.531 mmol). The Fmoc group was removed by nutating the resin in 5% DEA in DMF for 30 min at room temperature. The resin was washed with DMF and Fmoc-Phe-OH (617 mg, 1.593 mmol, 3 eq), HOBt (215 mg, 1.593 mmol, 3 eq), and DIC (249 µL, 1.593 mmol, 3 eq) were added and the reaction mixture nutated for 2 hr at room temperature. The resin was washed with DCM and DMF and the Fmoc group removed by nutating in 5% DEA in DMF for 30 min. The resin was washed with DCM and DMF and 4-pentynoic acid (156 mg, 1.593 mmol, 3eq), HOBt (215 mg, 1.593 mmol, 3 eq), and DIC (249 µL, 1.593 mmol, 3 eq) were added and the reaction mixture nutated overnight at room temperature. Intermediate **9** was cleaved from resin using 1% TFA in DCM for 15 min. Concentration with toluene *in vacuo* yielded a white crystalline solid. The crude was purified by HPLC (reverse phase $C_{18}$ column, $CH_3CN/H_2O$ 0.1% TFA, 10:90 to 80:20 over 9 CVs. Pure fractions were lyophilized and 160 mg (0.428 mmol, 80.6% yield) Intermediate **9** were afforded as a white powder.

## pent-4-ynamido-Phe-Lys(Boc)-CMK (10)

**Chemical structure 3.** pent-4-ynamido-Phe-Lys(Boc)-CMK.

Carboxylic acid **9** (127 mg, 0.34 mmol was converted to the chloromethylketone using the procedure described above. The crude material was purified by flash column chromatography (20% ethyl acetate in hexane -> 60% ethyl acetate in hexane), and pure fractions pooled to yield 25.6 mg (0.06 mmol, 19% yield) of white crystalline solid.

**Chemical structure 4.** pent-4-ynamido-Phe-Lys(Boc)-AOMK (GB-IA).

## GB-IA (4)

Intermediate **10** (25.6 mg, 0.05 mmol, 1 eq) was converted to the AOMK following the general procedure. The crude was purified by HPLC (reverse phase $C_{18}$ column, $CH_3CN/H_2O$ 0.1% TFA, 20:80 to 60:40 in x column volumes). Pure fractions were pooled and lyophilized. The lyophilized fractions were taken up in 50% TFA in DCM and stirred for 1 hr at room temperature. The reaction was concentrated with toluene *in vacuo* to yield 4.68 mg (9 µmol, 5.6% yield) of white crystalline solid, **GB-IA (4)**.

[1]H NMR (400 MHz, $CD_3OD/CDCl_3$ 1/1) δ 7.32 – 7.24 (m, 4H), 7.23 – 7.15 (m, 2H), 7.06 – 7.01 (m, 2H), 4.61 – 4.41 (m, 4H), 3.10 (dd, $J$ = 13.6, 8.4 Hz, 1H), 3.00 (dd, $J$ = 13.6, 7.4 Hz, 1H), 2.89 (t, $J$ = 7.4 Hz, 2H), 2.44 – 2.39 (m, 4H), 2.35 (s, 6H), 2.16 (t, $J$ = 2.2 Hz, 1H), 2.01 – 1.84 (m, 1H), 1.72 – 1.53 (m, 3H), 1.51 – 1.34 (m, 2H).

HRMS (ES+): $[M+H+]^+$ calculated for $C_{30}H_{37}N_3O_5$ expected mass 520.2811 found 520.2797. LCMS (ES+): retention time 6.55 min.

## Synthesis of ac-GB111 (5), az-GB (6), and GB111-PMK (2)

**Chemical structure 5.** Carboxybenzyl-Phe-Lys(Boc)-CMK.

## Cbz-Phe-Lys(Boc)-CMK (8)

Intermediate **7** (200 mg, 0.38 mmol, 1 eq) was converted to the chloromethyl ketone as described in the general procedure above. The crude was purified by flash column chromatography (20% ethyl acetate in hexane ->60% ethyl acetate in hexane), and pure fractions pooled to yield 150 mg (0.27 mmol, 70% yield) of white crystalline solid.

**Chemical structure 6.** caboxybenzyl-Phe-Lys-AOMK (GB111-NH₂).

## GB111-NH₂ (1)

Intermediate **8** (30 mg, 0.05 mmol) was converted to the acyloxymethylketone as described above in the general procedure. The crude was purified by HPLC (reverse phase $C_{18}$ column, $CH_3CN/H_2O$ 0.1% TFA, 20:80 to 60:40 over 25 min, 15 mL per minute. Pure fractions were lyophilized. Lyophilized fractions were taken up in 50% TFA in DCM and stirred for 30 min, after which the reaction mixture was concentrated with toluene *in vacuo* to yield 14.5 mg (25.26 µmol, 51%) **GB111-NH₂** as a white powder. Refer to Patent US2007/36725 A1 for previous synthetic scheme of Intermediates **7** and **8** and GB111-NH₂ and compound characterization.

**Chemical structure 7.** caboxybenzyl-Phe-Lys(Ac)-AOMK (ac-GB111).

## ac-GB111 (5)

GB111-NH$_2$ (**1**) (4.58 mg, 8.81 µmol,1 eq) was dissolved in anhydrous DCM. Triethylamine (1.35 µL, 9.69 µmol, 1.1 eq) was added and the reaction mixture stirred for 5 min before the addition of acetyl chloride (0.94 µL, 13.21 µmol, 1.5 eq). The mixture was stirred at room temperature for 30 min and then concentrated *in vacuo*. The crude was taken up in DMSO and purified by HPLC (reverse phase C$_{18}$ column, CH$_3$CN/H$_2$O 0.1% TFA, 20:80 to 50:50 over column volumes. Pure fractions were lyophilized to yield 0.45 mg (0.73 µmol, 8.2% yield) of white crystalline solid, **ac-GB111** (**5**).

$^1$H NMR (400 MHz, CD$_3$OD/CDCl$_3$ 1/1) δ 7.33 – 7.10 (m, 11H), 6.99 (d, $J$ = 7.4 Hz, 2H), 5.00 – 4.98 (m, 2H), 4.65 (s, 2H), 4.42 (dd, $J$ = 11.3, 6.3 Hz, 2H), 3.17 – 3.09 (m, 1H), 3.09 – 3.00 (m, 2H), 2.97 – 2.88 (m, 1H), 2.32 (s, 6H), 1.86 (s, 3H), 1.62 – 1.51 (m, 1H), 1.44 – 1.34 (m, 3H), 1.31 – 1.22 (m, 2H).

HRMS (ES+): [M+H+]$^+$ calculated for C$_{35}$H$_{41}$N$_3$O$_7$ expected mass 616.3023 found 616.3017. LCMS (ES+): retention time 8.08 min.

**Chemical structure 8.** caboxybenzyl-Phe-Lys(Az)-AOMK (az-GB).

## az-GB (6)

GB111-NH$_2$ (**1**) (2.2 mg, 3.83 µmol, 1 eq) was dissolved in anhydrous methanol. K$_2$CO$_3$ (1.68 mg, 12.2 µmol, 3 eq), imidazole-1-sulfonyl azide HCl (*Goddard-Borger and Stick, 2007*) (0.9 mg, 5.2 µmol, 1.36 eq), and Cu(II)SO$_4$ pentahydrate (0.0034 mg, 0.014 mmol, 0.003 eq) were added and the reaction mixture was stirred overnight at room temperature. The reaction mixture was concentrated *in vacuo*. The crude was taken up in DMSO and purified by HPLC (reverse phase C$_{18}$ column, CH$_3$CN/H$_2$O 0.1% TFA, 20:80 to 60:40 over column volumes. Pure fractions were lyophilized to yield 1.77 mg (2.95 µmol, 77% yield) of white crystalline solid, **az-GB** (**6**).

$^1$H NMR (500 MHz, CD$_3$OD/CDCl$_3$ 1/1) δ 7.37 – 7.19 (m, 11H), 7.06 (d, $J$ = 7.6 Hz, 2H), 5.08 (s, 2H), 4.71 – 4.60 (m, 2H), 4.51 – 4.44 (m, 2H), 3.25 (t, $J$ = 6.8 Hz, 2H), 3.11 (dd, $J$ = 13.6, 7.5 Hz, 1H), 2.99 (dd, $J$ = 13.6, 7.4 Hz, 1H), 2.39 (s, $J$ = 6.3 Hz, 6H), 1.99 – 1.87 (m, 1H), 1.68 – 1.51 (m, 3H), 1.51 – 1.32 (m, 2H).

HRMS (ES+): [M+H+]$^+$ calculated for C$_{33}$H$_{37}$N$_5$O$_6$ expected mass 600.2822 found 600.2818. LCMS (ES+): retention time 8.90 min.

**Chemical structure 9.** caboxybenzyl-Phe-Lys-PMK (GB111-PMK).

## GB111-PMK (2)

Potassium fluoride (15.56 mg, 0.27 mmol, 3 eq) and 2,3,5,6-tetrafluorophenol (16.3 mg, 0.1 mmol, 1.1 eq) were added to DMF and the reaction mixture stirred at 80°C for 10 min. Intermediate **10** (50.41 mg, 0.09 mmol, 1 eq) was taken up in DMF and added to the reaction mixture. This mixture was stirred for 2 hr at 80°C then concentrated *in vacuo*. The crude was taken up in DCM and purified by flash column chromatography (hexane -> 55% ethyl acetate in hexane). Pure fractions were pooled and concentrated *in vacuo*. This product was taken up in 50% TFA in DCM and stirred for 30 min, after which it was concentrated with toluene *in vacuo* to yield 35.3 mg **GB111-PMK** (**2**) (0.06 mmol, 65% yield) as a white crystalline solid.

$^1$H NMR (500 MHz, cd$_3$od) δ 7.35 – 7.19 (m, 10H), 7.16 – 7.06 (m, $J$ = 14.4, 8.7, 5.3 Hz, 1H), 5.09 – 4.94 (m, 2H), 4.81 – 4.68 (m, 2H), 4.55 – 4.44 (m, 1H), 4.39 – 4.32 (m, 1H), 3.09 – 2.91 (m, 2H), 2.85 (t, $J$ = 7.6 Hz, 2H), 1.95 – 1.74 (m, $J$ = 40.2 Hz, 1H), 1.68 – 1.49 (m, 3H), 1.48 – 1.34 (m, 2H).

HRMS (ES+): [M+H+]$^+$ calculated for C$_{30}$H$_{31}$F$_4$N$_3$O$_5$ expected mass 590.2278 found 590.2278. LCMS (ES+): retention time 7.04 min.

## Acknowledgements

We thank the Bogyo lab for helpful discussions regarding the direction of this project, M Child and S Ewald for comments on the manuscript, M Spitzer for comments and flow cytometry assistance, A Ondrus, T-H Lee, A Bhaduri, and E LaGory for assay advice and reagents. This work was supported by the National Science Foundation under grant DGE-114747 (to LES) and by the National Institutes of Health under grants R01 EB005011 and R01 CA179253 (to MB) and AI063302 and AI065359 (to DMM). EW is grateful for financial support from the Smith Family Foundation, the Damon Runyon Cancer Research Foundation (DRR-18-12) and Boston College.

## Additional information

### Funding

| Funder | Grant reference number | Author |
| --- | --- | --- |
| National Institutes of Health | R01 EB005011, R01 CA179253 | Matthew Bogyo |
| National Science Foundation | DGE-114747 | Laura E Sanman |
| Damon Runyon Cancer Research Foundation | DRR-18-12 | Eranthie Weerapana |
| National Institutes of Health | R01 AI063302, R01 AI065359 | Denise M Monack |

The funders had no role in study design, data collection and interpretation, or the decision to submit the work for publication.

## Author contributions

LES, Conception and design, Acquisition of data, Analysis and interpretation of data, Drafting or revising the article; YQ, NAE, TMN, Conception and design, Acquisition of data, Analysis and interpretation of data; WAvdL, Acquisition of data, Analysis and interpretation of data; DMM, EW, MB, Conception and design, Analysis and interpretation of data, Drafting or revising the article

## Author ORCIDs

Matthew Bogyo, ⓘ http://orcid.org/0000-0003-3753-4412

## Ethics

Animal experimentation: This work was approved under ABP protocol 1331 (Entitled Chemical probes to study host responses to bacterial pathogens) and APLAC protocol 18026. Primary cells were isolated from mouse bone marrow following strict accordance with the NIH guide for the care and use of laboratory animals. These protocols were reviewed and approved by the Environmental Health and Safety Department of Stanford University and the Institutional Animal Care and Use Committee of Stanford University, respectively.

## Additional files

### Supplementary files

• Supplementary file 1. Complete MudPIT dataset. BMDM from four different treatment conditions (see *Figure 2*) were lysed and lysates reacted with alkyne-biotin. Biotin-labeled proteins were enriched using streptavidin resin, digested off the resin, and bound proteins identified by mass spectrometry. Number of spectral counts from peptides from each identified protein are reported for each condition.

• Supplementary file 2. Complete MudPIT dataset normalized to endogenously biotinylated proteins. Data from *Supplementary file 1* was normalized based on relative abundance of endogenously biotinylated proteins across conditions. This data reports normalized spectral counts.

• Supplementary file 3. Hits from MudPIT analysis. A protein was selected as a 'hit' if, in *Supplementary file 2*, it had over 30 spectral counts in Condition 1, showed 80% competition for az-GB binding by GB111 (Condition 2), and showed <50% competition for az-GB binding by GB-IA (Condition 3). The proteins meeting these criteria and number of spectral counts in each condition are shown.

• Supplementary file 4. Compound Characterization and Spectra.

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
