## [Decision Letter]

Thank you for submitting your work entitled "Disruption of glycolytic flux is a signal for NLRP3 inflammasome formation and pyroptosis" for consideration by *eLife*. Your article has been favorably evaluated by Charles Sawyers (Senior editor) and three reviewers, one of whom is a member of our Board of Reviewing Editors.

The reviewers have discussed the reviews with one another and the Reviewing Editor has drafted this decision to help you prepare a revised submission.

The manuscript by Sanman and colleagues describes the discovery that disruption of glycolysis, either by chemical probes or pathogen infection, activates the NLRP3 inflammasome of macrophages. The initial findings were made using a small-molecule phenotypic screen, which furnished a covalent ligand that blocks GAPDH and enolase activity. This compound, as well as other GAPDH and enolase inhibitors, promoted NLRP3 inflammasome formation, and this effect was blocked by treating cells with the glycolytic end product pyruvate. Further studies indicate that reductions in NADH are important for promoting NLRP3 inflammasome activation. Similar effects were also observed in macrophages infected with *Salmonella typhimurium*, which appears to impair macrophage glycolysis by consuming host glucose.

The reviewers agreed that your manuscript reports novel findings of potentially sufficient general interest for eventual publication at *eLife*. However, two major concerns were raised that must be addressed with additional experiments.

1) Both Reviewers 1 and 2 expressed concern about the target identification experiments for GB111-NH_2_. More evidence to support how GB111-NH_2_ interacts with GAPDH and enolase-1 to apparently inhibit these proteins, for instance, would benefit the manuscript. Likewise, the use of additional inactive control probes could be helpful (e.g., probes that are structurally related to GB111-NH_2_, but not reactive, which would support that a covalent mechanism of target binding is required for the observed pharmacological effects).

2) A more thorough characterization of the effects of GB111-NH_2_ and comparison compounds on the metabolism and inflammatory responses (e.g., IL-1β secretion) of cells would strengthen the paper, as noted in more detail by Reviewer 3.

Reviewer 1:

1) The target identification and validation for GB111-NH_2_ was difficult for this Reviewer to follow. More details on how the final candidate target list ([Supplementary-material SD3-data]) was generated would be helpful (e.g., what filter were used to designate these proteins as targets over other proteins identified in their proteomic experiments). Also, the data in Figure 4 are confusing – this Reviewer had difficulty noting any substantive change in az-GB labeling of GAPDH in the GB111-NH_2_ competition lane. Likewise, why is there a background signal in the DMSO lanes for both enzymes? Perhaps this relates to non-specific reactivity of the TAMRA-alkyne tag with the proteins, since click reactions run in the orientation used by the authors are known to exhibit some background reactivity. This point should be clarified. The authors should also consider performing a quantitative, concentration-dependent analysis of GB111-NH_2_ blockade of az-GB labeling of GAPDH and Eno1, which might better convince readers that these chemical probes are competitively binding/reacting with the same site on each enzyme.

2) Somewhat related to point 1, the authors state in the subsection “Inhibition of glycolytic enzymes activates the NLRP3 inflammasome and induces 139 pyroptosis” that both GAPDH and enolase have "key catalytic site cysteine residues". This Reviewer knows that GAPDH uses a conserved cysteine for catalysis, but was not aware that enolase depends on cysteine residues for activity. Could the authors clarify this statement as relates to enolase and cite relevant papers showing the importance of one or more active-site cysteines in enolase. This is important, since the authors do not appear to directly identify the site of GB111-NH_2_ reactivity in either enzyme.

Reviewer 2:

Although not explicitly stated in the manuscript, the method used to identify cellular targets of GB111-NH_2_ is heavily biased towards the identification of proteins that are covalently modified by the small molecule. Therefore, potential cellular targets making non-covalent interactions with GB111-NH_2_ are likely missed. This should be stated clearly in the text, and is a limitation of the analysis.

The cellular effects of GB111-NH_2_ are compared against those of published GADPH and α-enolase inhibitors. It is unclear to me if these compounds are selective and properly validated- the data in the references I can find is quite limited. Would be good to clarify this with additional data or text.

The 'inactive compound' still appears to be active against α-enolase (only 2-3 fold less inhibition at 10 µM). Is dual inhibition of GAPDH and α-enolase required for inflammatory signaling or is it only inhibition of GAPDH that is relevant? If we assume that the published ENOblock compound is selective for α-enolase, then this should not be the case.

Does GB111-NH_2_ react with unique sites on GAPDH/α-enolase? The SAR data shown suggests that it does, however there is no mutational analysis of the proteins to confirm. Furthermore, do the published inhibitors used in the study bind/react with similar sites?

Finally, mapping the binding site of GB111-NH_2_ would potentially enable identification of resistance mutations, which could be used to convincingly show that GAPDH/α-enolase are indeed the phenotypically relevant targets of the compound.

Reviewer 3:

Figure 5 – measurement of NAD/NADH levels – LPS/GB111 vs. LPS/Nigericin and showing that nigericin does not behave like GB111 in terms of altering NAD/NADH ratios – this is a useful control but are kinetics/extent of inflammasome activation in GB111 vs. nigericin treated cells the same under these conditions? Measurement of LDH release or PI uptake by these cells and showing that they are equivalent is important for this interpretation to be correct. LPS + Enoblock does not seem to do the same thing as LPS + GB111 or LPS + KA (Figure 5), suggesting that blocking α-enolase alone, while inducing inflammasome activation (by IL-1β secretion) does not do so in the same way as GB111.

Addition of downstream metabolites to GB111-treated cells prevented inflammasome activation providing a nice test of the authors' hypothesis. A prediction is that these metabolites should also reverse alteration of the NAD/NADH ratio by GB111. Is this the case? If the NAD/NADH ratio is the thing being sensed by the cell as a trigger of inflammasome activation, is it possible to directly alter these ratios in primed cells to mimic the effect of GB111, or conversely to override the effect of GB111?

Figure 7 – lack of inhibition of GB111-induced inflammasome activation by extracellular potassium is quite interesting, but it would be important to have an additional readout besides just% of cells with ASC foci – LDH release or IL-1β secretion – 5% is quite low, and although the background tends to be very low as well, it isn't much above noise, so having an additional way of confirming this is important.

Figure 9 – the levels of IL-1β secretion that are being measured in Figure 9 are low relative to what is typically seen. While there are other data that support the conclusion that disruption of glycolysis by *Salmonella* induces NLRP3 inflammasome previous literature (for example Broz et al. 2014) suggests that there should be quite a bit more IL-1β secretion under these conditions. Moreover, while it is fair to say that pyruvate reduces *Salmonella*-induced inflammasome activation, a drop from 80 to 40 pg/mL IL-1β or a drop from 30% to 25% LDH does not seem exceptionally convincing – since all of the inflammasome activation that *Salmonella* is inducing under these conditions depends on NLRP3, what accounts for the pyruvate-independent inflammasome activation in response to *Salmonella*? Presumably this is due to caspase-11. At the very least this discrepancy between the effect of pyruvate on *Salmonella*-induced inflammasome activation and GB111 induced inflammasome activation should be discussed. A further test of whether *Salmonella* induces NLRP3 inflammasome activation by altering the NAD/NADH ratio is whether adding pyruvate or glutamine to *Salmonella*-infected cells prevents this NLRP3 inflammasome activation.

Figure 1—figure supplement 1 – this is a poor quality western blot that should really be cleaned up for publication purposes. The relevant band is hard to distinguish from the background in the western blot. Even though this is a supplemental figure it should still be done to the same standard as a main text figure.

Figure 1—figure supplement 2 – the title of this figure is that GB111 doesn't impair secretion of inflammasome-independent pro-inflammatory cytokines, but it has this effect only for TNF – it DOES inhibit IL-6 in a dose-dependent way. The explanation is likely related to the difference in kinetics of IL-6 and TNF transcription – since the priming was only done for 3 hours, it is likely that at higher concentrations of GB111 more of the cells are dying more quickly, leading to the appearance of an effect on IL-6, but not TNF since TNF is a primary response gene and much more rapidly expressed. This is testable with a longer period of LPS priming. This should either be done or the title of the figure legend should be changed to appropriately reflect the data.

---

## [Author Response]

*The reviewers agreed that your manuscript reports novel findings of potentially sufficient general interest for eventual publication at eLife. However, two major concerns were raised that must be addressed with additional experiments. 1) Both Reviewers 1 and 2 expressed concern about the target identification experiments for GB111-NH_2_. More evidence to support how GB111-NH_2_ interacts with GAPDH and enolase-1 to apparently inhibit these proteins, for instance, would benefit the manuscript. Likewise, the use of additional inactive control probes could be helpful (e.g., probes that are structurally related to GB111-NH_2_, but not reactive, which would support that a covalent mechanism of target binding is required for the observed pharmacological effects).*We conducted several additional studies to further demonstrate how GB111-NH_2_ interacts with both GAPDH and α-enolase and added data from these studies to the manuscript.

We agree that the GAPDH labeling by az-GB was initially hard to interpret (revised manuscript Figure 4—figure supplement 1). We optimized buffer conditions and now observe much cleaner labeling. Using these conditions, we performed a dose-response pretreatment of GB111-NH_2_ before labeling with the az-GB probe. We observe that GB111-NH_2_ inhibits az-GB binding to GAPDH and α-enolase in a dose-dependent manner, indicating that it likely binds to a discrete site on both enzymes. Compound 6, which contains a more reactive electrophile than GB111-NH_2_ and more potently induces IL-1β secretion in BMDM (Figure 2), also competed with az-GB for binding to both GAPDH and α-enolase. Importantly, this compound showed competition at lower concentrations than GB111-NH_2_, consistent with it having greater potency for the two targets. Furthermore, as suggested by the reviewers, we more carefully evaluated the two negative control compound. This includes compound 2 which lacks the electrophile and GB-IA which is identical to GB111 but lacks a carboxybenzyl cap. Both of these compounds did not induce IL-1β release and we show now also do not compete for az-GB binding to GAPDH or α–enolase (revised manuscript Figure 4). Collectively, these data strengthen our correlation between in vitro target-binding and cellular effects.

Our initial observation in our original manuscript that the cysteine alkylating compound N-ethylmaleimide blocked az-GB binding to GAPDH and α-enolase (revised manuscript Figure 4—figure supplement 1), suggested that GB111-NH_2_ covalently binds to reactive cysteine residues on GAPDH and α-enolase. To further explore this mechanism of compound binding, we performed labeling experiments on purified GAPDH and α-enolase with iodoacetamide fluorescein (IAF), which labels reactive cysteine residues. Our collaborator in this study has previously shown (Weerapana et al., Nature 2010) that iodoacetamide preferentially labels the active site Cys 152 of GAPDH and Cys 388 of α-enolase. Notably, chemical modification of Cys 388 is deleterious to α-enolase activity (Ishii et al., Chem Res Toxicol 2004). Our new data (revised manuscript Figure 4) shows that IAF labels both GAPDH and α-enolase and is competed by both N-ethylmaleimide and by GB111-NH_2_. Together, these results indicate that GB111-NH_2_ binds to the same active site cysteines as IA and support a mechanism in which GB111-NH2 function by blocking activity of these enzyme targets by direct binding to active site cysteine residues.

To further support a covalent mechanism of target binding, we conducted experiments in which we pretreated GAPDH and α-enolase with GB111-NH_2_ for increasing periods of time. Longer pretreatment resulted in increased inhibition of both GAPDH and α-enolase (revised manuscript Figure 4), which indicates that GB111-NH_2_ is a covalent inhibitor, again consistent with a mechanism in which inhibition results from covalent modification of the active site cysteines.

Taken together, these data support the hypothesis that GB111-NH_2_ covalently binds to reactive cysteine residues on both GAPDH and α-enolase and identifies target cysteines that GB111-NH_2_ may bind. Furthermore, these data strengthen the correlation between the binding of our GB111-NH_2_ analogs to GAPDH and α-enolase and their ability to induce IL-1β secretion in cells.

Finally, while we agree rigorous analysis of target binding is important for any small molecule, we would like to point out that the major diving focus of this paper is not to identify a new class of selective inhibitors of GAPDH or α-enolase. Rather, we use these tools to uncover a previously unknown pathway for activation of pyroptosis that is relevant to pathogen infection. By showing this result using many different classes of small molecule and pathogen stimuli we feel that we have made the point very clearly. Finally, by rescuing the phenotype induced by both the small molecule stimuli as well as the pathogen using downstream components of glycolysis, we confirm that changes in glycolytic flux are in fact responsible for the induction of pyroptosis, thus making extensive characterization of all possible targets of the original small molecule relatively unimportant to the conclusions of the paper.

*2) A more thorough characterization of the effects of GB111-NH_2_ and comparison compounds on the metabolism and inflammatory responses (e.g., IL-1β secretion) of cells would strengthen the paper, as noted in more detail by Reviewer 3.*As suggested by the reviewers, we identified a concentration of nigericin (1 μM) which has a similar timecourse of inflammasome formation/pyroptosis to 10 μM GB111-NH_2_. We analyzed NADH levels (alongside LDH release and IL-1β secretion) and found that NADH levels are reduced by treatment with GB111-NH_2_, but that the nigericin controls do not show this reduction in NADH levels (see Figure 5—figure supplement 2). We think that these data more convincingly indicate that changes in metabolism observed upon GB111-NH_2_ treatment are indeed due to its proposed mechanism of action rather than a side effect of inflammasome formation and/or cell death.

We agree with the Reviewer’s conclusion that LPS+Enoblock has a somewhat different metabolic signature from LPS+GB111 and LPS+KA, specifically that NAD+/NADH ratio is not altered in BMDM by Enoblock treatment. Therefore, it is possible that a) all glycolytic inhibitors converge on a different secondary signal than NAD+/NADH that creates mitochondrial ROS and activates the inflammasome or, b) Enoblock activates the inflammasome through a distinct mechanism. We addressed these possibilities in the text as they are important avenues for future studies.

*Reviewer 1: 1) The target identification and validation for GB111-NH_2_ was difficult for this Reviewer to follow. More details on how the final candidate target list ([Supplementary-material SD3-data]) was generated would be helpful (e.g., what filter were used to designate these proteins as targets over other proteins identified in their proteomic experiments). Also, the data in Figure 4 are confusing – this Reviewer had difficulty noting any substantive change in az-GB labeling of GAPDH in the GB111-NH_2_ competition lane. Likewise, why is there a background signal in the DMSO lanes for both enzymes? Perhaps this relates to non-specific reactivity of the TAMRA-alkyne tag with the proteins, since click reactions run in the orientation used by the authors are known to exhibit some background reactivity. This point should be clarified. The authors should also consider performing a quantitative, concentration-dependent analysis of GB111-NH_2_ blockade of az-GB labeling of GAPDH and Eno1, which might better convince readers that these chemical probes are competitively binding/reacting with the same site on each enzyme.*The Reviewer is correct regarding the background signal in the DMSO lanes – we believe that the background labeling we see is nonspecific because alkynes can function as cysteine-targeting electrophiles (Ekkebus et al., JACS 2013) and both GAPDH and α-enolase contain reactive cysteine residues. We changed the buffer to one in which nonspecific interaction of the TAMRA-alkyne with α-enolase and GAPDH is reduced to allow clearer interpretation of our results (Figure 4). We will address this in the text to clarify for the readers.

We did experiments looking at competition of GB111-NH_2_ and analogs for az-GB binding to both GAPDH and α-enolase over several concentrations, as the reviewer suggested. We observe dose-dependent inhibition of az-GB binding to both GAPDH and α-enolase by GB111-NH_2_. Compound 6, which contains a more reactive electrophile than GB111-NH_2_ and more potently induces IL-1β secretion in BMDM (Figure 2), also competed az-GB binding to both GAPDH and α-enolase more potently than GB111-NH_2_. Importantly, both the GB111-NH_2_ analog that lacks an electrophile (compound 2) and the GB111 analog that lacks a carboxybenzyl cap (GB-IA) (both of which did not induce IL-1β secretion in BMDM) did not compete for az-GB binding to GAPDH and α–enolase (revised manuscript Figure 4).

In addition, we conducted experiments where we pretreated GAPDH and α-enolase with GB111-NH_2_ for increasing amounts of time and then measured activity. Longer pretreatment time resulted in less enzyme activity (Figure 4), which is a characteristic of covalent inhibitors.

*2) Somewhat related to point 1, the authors state in the subsection “Inhibition of glycolytic enzymes activates the NLRP3 inflammasome and induces 139 pyroptosis” that both GAPDH and enolase have "key catalytic site cysteine residues". This Reviewer knows that GAPDH uses a conserved cysteine for catalysis, but was not aware that enolase depends on cysteine residues for activity. Could the authors clarify this statement as relates to enolase and cite relevant papers showing the importance of one or more active-site cysteines in enolase. This is important, since the authors do not appear to directly identify the site of GB111-NH2 reactivity in either enzyme.*We agree that this statement is a bit confusing and will clarify both here and in the main text of the paper. As the Reviewer notes, GAPDH has an active site cysteine. Our collaborator (Weerapana et al., Nature 2010) has identified a reactive cysteine in α-enolase that can be labeled with iodoacetamide. Based on the crystal structure of human α-enolase (Kang et al., Acta Crystallogr Sect D 2008), this cysteine, Cys 388 is close to the entrance to the enzyme’s substrate-binding site. Furthermore, it has previously been shown that modification of this cysteine residue results in a loss of activity (Ishii et al., Chem Res Toxicol 2004).

We did an experiment to confirm that GAPDH and α-enolase contain reactive cysteines by labeling with the reactive cysteine probe iodoacetamide-fluorescein (IAF). Both label with IAF and labeling can be blocked by incubation with the cysteine-alkylating compound N-ethylmaleimide (NEM). We observe that GB111-NH_2_ pretreatment results in reduced iodoacetamide fluorescein labeling (Figure 4), indicating that GB111-NH_2_ binds to these reactive cysteines.

*Reviewer 2: Although not explicitly stated in the manuscript, the method used to identify cellular targets of GB111-NH_2_ is heavily biased towards the identification of proteins that are covalently modified by the small molecule. Therefore, potential cellular targets making non-covalent interactions with GB111-NH_2_ are likely missed. This should be stated clearly in the text, and is a limitation of the analysis.*This will be stated in the text. Based on data with our non-reactive analog of GB111-NH_2_, we did not think that non-covalent interactions are important for the phenotype under study, but we will state this clearly anyway.

*The cellular effects of GB111-NH_2_ are compared against those of published GADPH and α-enolase inhibitors. It is unclear to me if these compounds are selective and properly validated- the data in the references I can find is quite limited. Would be good to clarify this with additional data or text.*The relevant papers in which specificity characterization was reported have been cited. Specifically, specificity of KA was determined amongst enzymes in the glycolytic pathway in Endo et al., J Antibiotics 1985 (cited in our manuscript). Proteomic analysis has not been undertaken but its effects in different species correlate with the in vitrosensitivity of those species GAPDH isoforms (indirect genetic evidence of its effects). When EB was first reported (ACS Chem Biol 2013), the authors performed an affinity enrichment of all binding partners of EB. The proteins that they identified were all enolase isoforms (Jung et al., ACS Chem Biol 2013).

*The 'inactive compound' still appears to be active against α-enolase (only 2-3 fold less inhibition at 10 µM). Is dual inhibition of GAPDH and α-enolase required for inflammatory signaling or is it only inhibition of GAPDH that is relevant? If we assume that the published ENOblock compound is selective for α-enolase, then this should not be the case.*Our data suggest that inhibition of either GAPDH or α-enolase is capable of activating the inflammasome. GAPDH inhibition may drive the phenotype more than α-enolase inhibition, based on the observation that the metabolic signature of GB111-NH_2_ is more similar to KA. However, the increase in potency from GB111-NH_2_ relative to KA or EB alone may be due to a synergistic effect of blocking both. Therefore, we do not think that the in vitro activity that the ‘inactive compound’ exhibits towards -enolase is sufficient to cause it to have activity in cells, especially if α-enolase inhibition is less of a driver of the biological phenotype.

*Does GB111-NH_2_ react with unique sites on GAPDH/α-enolase? The SAR data shown suggests that it does, however*α

*there is no mutational analysis of the proteins to confirm. Furthermore, do the published inhibitors used in the study bind/react with similar sites?*KA binds to the catalytic cysteine of GAPDH. The binding site of EB on α-enolase has not been characterized. We did a competition labeling study of GAPDH and α-enolase with 10 μM iodoacetamide-fluorescein (IAF) (Figure 4). According to proteomic data previously generated by the authors (Weerapana et al., Nature 2010), 10 μM iodoacetamide will predominantly label the active site Cys 152 of GAPDH and Cys 388 of α-enolase (a cysteine flanking the substrate-binding pocket of α-enolase). We see that IAF binding to both GAPDH and α-enolase is competed by GB111-NH_2_ pretreatment, indicating that GB111-NH_2_ also binds to these cysteines.

*Finally, mapping the binding site of GB111-NH_2_ would potentially enable identification of resistance mutations, which could be used to convincingly show that GAPDH/α-enolase are indeed the phenotypically relevant targets of the compound.*An interesting study, but beyond the scope of this paper. The feasibility of this study is also questionable because it has previously been shown that expression of GAPDH mutants that lack activity is cytotoxic (Yogalingam et al., JBC 2013), and our studies are being conducted in BMDM which do not live long enough in culture to generate resistance mutants.

*Reviewer 3:Figure 5 – measurement of NAD/NADH levels – LPS/GB111 vs. LPS/Nigericin and showing that nigericin does not behave like GB111 in terms of altering NAD/NADH ratios – this is a useful control but are kinetics/extent of inflammasome activation in GB111 vs. nigericin treated cells the same under these conditions? Measurement of LDH release or PI uptake by these cells and showing that they are equivalent is important for this interpretation to be correct. LPS + Enoblock does not seem to do the same thing as LPS + GB111 or LPS + KA (Figure 5), suggesting that blocking α-enolase alone, while inducing inflammasome activation (by IL-1β secretion) does not do so in the same way as GB111.*We have done a more careful titration of nigericin to find a concentration at which the kinetics of inflammasome activation and cell death are similar to those in GB111-NH_2_-treated BMDM. At this concentration of nigericin (1 μM), we see a decrease in NADH production upon GB111-NH_2_ treatment but do not observe a decrease in NADH production in the nigericin-treated cells (Figure 5—figure supplement 1).

The α-enolase inhibitor does not have an identical metabolic signature to the GAPDH inhibitor or GB111-NH_2_ – though lactate production is impaired, NAD+/NADH is not significantly altered. This either indicates that the α-enolase inhibitor is working through a distinct mechanism from glycolytic inhibition or that NAD+/NADH, while predictive of inflammasome formation, is not the universal signal sensed downstream of glycolytic disruption. To prevent confusion for future studies, this will be directly discussed in the text.

*Addition of downstream metabolites to GB111-treated cells prevented inflammasome activation providing a nice test of the authors' hypothesis. A prediction is that these metabolites should also reverse alteration of the NAD/NADH ratio by GB111. Is this the case? If the NAD/NADH ratio is the thing being sensed by the cell as a trigger of inflammasome activation, is it possible to directly alter these ratios in primed cells to mimic the effect of GB111, or conversely to override the effect of GB111?*These metabolites do reverse the NAD+/NADH ratio defect observed upon GB111 treatment – see Figure 6. We do override this by adding rotenone (see Figure 7), where we see a reversal of the NAD+/NADH ratio caused by GB111 and then see complete suppression of inflammasome formation induced by GB111.

*Figure 7 – lack of inhibition of GB111-induced inflammasome activation by extracellular potassium is quite interesting, but it would be important to have an additional readout besides just% of cells with ASC foci – LDH release or IL-1β secretion – 5% is quite low, and although the background tends to be very low as well, it isn't much above noise, so having an additional way of confirming this is important.*We think that this is also an interesting point so we also looked at LDH release under these conditions. We also observe that high extracellular K^+^ blocks nigericin-induced LDH release but does not have a significant effect on GB111-induced LDH release (Figure 7—figure supplement 2).

*Figure 9 – the levels of IL-1β secretion that are being measured in Figure 9 are low relative to what is typically seen. While there are other data that support the conclusion that disruption of glycolysis by Salmonella induces NLRP3 inflammasome previous literature (for example Broz* et al.

*2014) suggests that there should be quite a bit more IL-1β secretion under these conditions. Moreover, while it is fair to say that pyruvate reduces Salmonella-induced inflammasome activation, a drop from 80 to 40 pg/mL IL-1β or a drop from 30% to 25% LDH does not seem exceptionally convincing – since all of the inflammasome activation that Salmonella is inducing under these conditions depends on NLRP3, what accounts for the pyruvate-independent inflammasome activation in response to Salmonella? Presumably this is due to caspase-11. At the very least this discrepancy between the effect of pyruvate on Salmonella-induced inflammasome activation and GB111 induced inflammasome activation should be discussed. A further test of whether Salmonella induces NLRP3 inflammasome activation by altering the NAD/NADH ratio is whether adding pyruvate or glutamine to Salmonella-infected cells prevents this NLRP3 inflammasome activation.*The difference could be due to caspase-11. Another explanation is that pyruvate supplementation is not completely blocking NLRP3 inflammasome formation because it is actively used – it is not a genetic block. However, it is true that pyruvate more effectively blocks GB111-NH_2_-induced NLRP3 inflammasome formation than *Salmonella typhimurium*-induced NLRP3 inflammasome formation, and we have added text to point this out and discuss both potential reasons for this difference. As a note, we do observe that cells infected with *Salmonella*, when supplemented with pyruvate, regain their ability to turn over NAD+ (Figure 9).

*Figure 1—figure supplement 1 – this is a poor quality western blot that should really be cleaned up for publication purposes. The relevant band is hard to distinguish from the background in the western blot. Even though this is a supplemental figure it should still be done to the same standard as a main text figure.*While we agree that the western blot is not ideal, we have not be able to get any significantly better results with the antibody we have. We think that the results are still clear in that we consistently see the appearance of the processed IL-1β mature form in cells at concentrations of GB111-NH_2_ where we get induction of pyroptosis and accumulation of IL-1b signals by ELISA.

Figure 1—figure supplement 2 – the title of this figure is that GB111 doesn't impair secretion of inflammasome-independent pro-inflammatory cytokines, but it has this effect only for TNF – it DOES inhibit IL-6 in a dose-dependent way. The explanation is likely related to the difference in kinetics of IL-6 and TNF transcription – since the priming was only done for 3 hours, it is likely that at higher concentrations of GB111 more of the cells are dying more quickly, leading to the appearance of an effect on IL-6, but not TNF since TNF is a primary response gene and much more rapidly expressed. This is testable with a longer period of LPS priming. This should either be done or the title of the figure legend should be changed to appropriately reflect the data.

We agree with the reviewer’s explanation of the IL-6 data, given published time course data of IL-6 secretion after LPS priming (Bjorkbacka et al. Physiol. Genomics 2004). We changed the title of the figure legend accordingly.